# Discovery of a drug candidate for GLIS3-associated diabetes

Sadaf Amin[1,2], Brandoch Cook[2], Ting Zhou[2], Zaniar Ghazizadeh[2], Raphael Lis[3], Tuo Zhang [4], Mona Khalaj[1,5], Miguel Crespo[2], Manuradhi Perera[2], Jenny Zhaoying Xiang[4], Zengrong Zhu[6], Mark Tomishima[6,7], Chengyang Liu[8], Ali Naji[8], Todd Evans[2], Danwei Huangfu [6] & Shuibing Chen[1,2,9]

GLIS3 mutations are associated with type 1, type 2, and neonatal diabetes, reflecting a key function for this gene in pancreatic β-cell biology. Previous attempts to recapitulate disease-relevant phenotypes in $GLIS3^{-/-}$ β-like cells have been unsuccessful. Here, we develop a "minimal component" protocol to generate late-stage pancreatic progenitors (PP2) that differentiate to mono-hormonal glucose-responding β-like (PP2-β) cells. Using this differentiation platform, we discover that $GLIS3^{-/-}$ hESCs show impaired differentiation, with significant death of PP2 and PP2-β cells, without impacting the total endocrine pool. Furthermore, we perform a high-content chemical screen and identify a drug candidate that rescues mutant GLIS3-associated β-cell death both in vitro and in vivo. Finally, we discovered that loss of GLIS3 causes β-cell death, by activating the TGFβ pathway. This study establishes an optimized directed differentiation protocol for modeling human β-cell disease and identifies a drug candidate for treating a broad range of GLIS3-associated diabetic patients.

[1] Weill Graduate School of Medical Sciences of Cornell University, 1300 York Avenue, New York, NY 10065, USA. [2] Department of Surgery, 1300 York Avenue, New York, NY 10065, USA. [3] Division of Regenerative Medicine, Department of Medicine, Ansary Stem Cell Institute, 1300 York Avenue, New York, NY 10065, USA. [4] Genomics Resources Core Facility, 1300 York Avenue, New York, NY 10065, USA. [5] Human Oncology and Pathogenesis Program, Memorial Sloan Kettering Cancer Center, New York, NY 10065, USA. [6] Developmental Biology Program, Sloan Kettering Institute, 1275 York Avenue, New York, NY 10065, USA. [7] SKI Stem Cell Research Facility, Sloan Kettering Institute, New York, NY 10065, USA. [8] Department of Surgery, University of Pennsylvania School of Medicine, Philadelphia, PA 19104, USA. [9] Department of Biochemistry, Weill Cornell Medicine, 1300 York Avenue, New York, NY 10065, USA. Correspondence and requests for materials should be addressed to D.H. (email: huangfud@mskcc.org) or to S.C. (email: shc2034@med.cornell.edu)

Candidate gene and genome-wide association studies (GWAS) have identified ~150 susceptibility loci for type 1 (T1D) and type 2 (T2D) diabetes. Of the genes identified so far, GLIS3 is the only one (other than insulin) associated with T1D[1–3], T2D[4–7], and, in addition, neonatal diabetes (ND)[8]. During mouse development, PDX1[+] pancreatic progenitors appear around embryonic day (E) E8.5; at E11.5, a small subset gives rise to mostly poly-hormonal endocrine cells commonly referred to as "primary transition" endocrine cells that likely do not contribute to the mature β-cell pool[9]. At E14.5, the "secondary transition" begins with extensive differentiation and emergence of mono-hormonal β-cells[10]. Glis3 begins to be expressed only in the secondary transition stage, is continually expressed in pancreatic β-cells and ductal cells[11], and plays a critical role in endocrine development[12]. In Glis3-deficient mice all subtypes of endocrine cells, especially β and δ cells[12], are significantly reduced, causing ND[13,14]. Glis3 is also essential for compensatory β-cell proliferation in adult mice[15]. The absence or decreased expression of Glis3 predisposes the mice to T2D[15,16]. In addition Glis3 mutations in non-obese diabetic (NOD) mice have been shown to underlie β-cell fragility and susceptibility to T1D[17]. However, the role of GLIS3 in human pancreatic development and human β-cells remains unclear.

Human pluripotent stem cells (hPSCs) have provided robust platforms to recapitulate pancreatic β-cell defects in diabetes, including maturity-onset diabetes of the young[18] and neonatal diabetes[19–21]. Recently, we used an isogenic hESC differentiation platform to evaluate the role of T2D-associated genes in pancreatic β-cell function and survival in disease conditions[22]. However, our initial attempt using isogenic GLIS3[−/−] hESCs failed to recapitulate the defects observed in Glis3[−/−] mice[20]. This raised the question whether GLIS3 plays different roles in mouse and human or whether the current differentiation strategy is not optimal to model GLIS3-related pancreatic β-cell defects. To distinguish between these possibilities, we monitored GLIS3 mRNA in hESC-derived pancreatic progenitors and INS[+] cells and found that the expression GLIS3 mRNA is undetectable, suggesting that the previous protocol[23] failed to efficiently generate the disease-relevant cells[20]. Here, we describe an optimized strategy to efficiently derive GLIS3[+] late-stage pancreatic progenitors (PP2), which give rise to mono-hormonal pancreatic β-cells (PP2-β cells). We use this platform to determine the role of GLIS3 in human pancreatic β-cell generation and survival, and to identify a lead hit drug candidate for treating the broad range of human patients who suffer from GLIS3-associated diabetes.

## Results

**Derivation of late-stage PPs that give rise to mono-hormonal cells.** Lacking an effective antibody for analyzing GLIS3 by immunocytochemistry, we used an indirect functional readout to identify conditions that promote the generation of PP2 cells with the capacity to differentiate into mono-hormonal insulin-expressing β-like cells. To compare conditions, INS[GFP/W] HES3 hESCs were differentiated to the early-stage pancreatic progenitors (PP1 at day 9/D9, Fig. 1a, Supplementary Table 1), giving rise to a pool that contains around 75–90% PDX1[+] cells (Supplementary Fig. 1a, b). The PP1 cells can differentiate into INS[+] (PP1-β) cells when cultured for seven additional days in basal differentiation medium (DMEM B27, Fig. 1a). However, the derived PP1-β cells are mostly poly-hormonal (comprising a population of 60–70% poly-hormonal and 30–40% mono-hormonal INS[+] cells), which represent the cells from older protocols[23,24] (Fig. 1d–f). The INS-GFP[+] PP1-β cells do not express detectable levels of GLIS3 by RT-PCR (Supplementary Fig. 1d). Poly-hormonal INS-GFP[+] PP1-β were previously shown to

differentiate mostly to α-cells when transplanted in vivo[25], which suggests that their identity is closer to the primary transition cells in mouse development. We performed a pilot screen to establish a strategy to promote the generation of PP2 cells that give rise to mono-hormonal INS[+] cells. Among 14 different culture conditions, we found one that consistently generates the highest percentage of PP2-derived insulin[+]/glucagon[−]somatostatin[−] (INS[+]/GCG[−]SST[−]) cells for the total INS[+] population (Supplementary Fig. 1c). This was achieved with "PP extension medium" containing 2 μM RA, 200 nM LDN193189, 0.25 μM SANT1, 10 ng/mL EGF, and 10 ng/mL FGF2. After 14 days of culture (from day 9 to day 23) in PP extension medium, more than 90% of the cells expressed PDX1 at day 23/D23_L (Supplementary Fig. 1b). Compared with PP1, PP2 cells express higher levels of late trunk PP markers, including NKX6.1 and NEUROD1 as indicated by qRT-PCR assays (Fig. 1b, Supplementary Table 2) and RNA-seq profiling (Fig. 1c). More importantly, after 7 days of differentiation, 85–95% of INS[+] cells derived from PP2 are mono-hormonal, expressing insulin, but not glucagon (Fig. 1d–f), somatostatin, or ghrelin (Supplementary Fig. 1g–j). In contrast, only 30–40% of INS[+] PP1-β cells are mono-hormonal (Fig. 1d–f and Supplementary Fig. 1h, j). In addition, PP2-β cells co-express mature β-cell markers (Fig. 1g, h), including PDX1 (97.3%), NKX2.2 (98.8%), PAX6 (86.0%), ISL1 (91.8%), and NKX6.1 (50.8%), and they also express UCN3 (63.6%), a mature β-cell marker[26] that was not reported as expressed using any of three previously published protocols[27–29]. We also looked at the mRNA expression of UCN3 and another β-cell marker MAFA. There is no significant difference in UCN3 and MAFA expression detected between PP2-β cells and primary human islets (Supplementary Fig. 1n). However, it is worth noting that big variation of UCN3 and MAFA expression was detected among different batches of human islets. RNA-seq profiling validated the downregulation of α-cell markers, ARX and glucagon, and other non-β-cell hormones in the purified INS-GFP[+] PP2-β cells, further confirming their mono-hormonal identity. Moreover, mature pancreatic β-cell markers are upregulated in the PP2-β INS[+] cells (Fig. 1h). Gene set enrichment analysis (GSEA) indicates that PP2-β cells closely resemble primary adult human β-cells. An upregulated gene set, comprising 1000 genes that are more highly expressed in adult human β-cells (~5-fold), is enriched in PP2-β cells; the downregulated gene set, comprising 1000 genes expressed at lower levels (~4-fold) in adult human β-cells, is enriched in PP1-β cells (Fig. 1i). Strikingly, PP2 and PP2-β cells express high levels of GLIS3 RNA, whereas GLIS3 transcripts are expressed at much lower levels in both PP1 and PP1-β cells (Fig. 1b–j, Supplementary Fig. 1d–f ). Compared to PP2 cells, the D17 cells generated using the previous protocol show limited GLIS3 expression, which explains why our previous studies did not detect defects of GLIS3[−/−] hESCs[20] (Supplementary Fig. 1e). Finally, cells at D30_L release insulin in response to stimulation with 20 mM glucose (Fig. 1k and Supplementary Fig. 1k) or other β-cell secretagogues, including 35 mM KCl, 30 μM Forskolin, or 10 mM Arginine (Fig. 1l). Using this strategy, mono-hormonal INS[+] cells were also derived with similar efficiency from HUES8 and H1 lines, demonstrating that that this differentiation strategy is not hESC line-dependent (Supplementary Fig. 1l and Supplementary Table 3). Also, the majority of α-like, δ-like, and ε-like cells at D30_L are mono-hormonal (Supplementary Fig. 1m).

**The differentiation of GLIS3[−/−] hESCs to β-like cells is impaired.** We used this improved differentiation protocol to evaluate the role of GLIS3 in human pancreatic development and generation of pancreatic β-like cells. To create biallelic GLIS3 mutant hESC lines, indel mutations were induced in INS[GFP/W]

HES3 cells using an sgRNA targeting exon 3 of *GLIS3* gene, predicted to impact all splice variants (Supplementary Fig. 2a and Supplementary Tables 4–6). The indel mutations were confirmed by Sanger sequencing (Supplementary Fig. 2b, c) and are

predicted to create an early frameshift and generate null alleles. To account for possible clonal variation, we analyzed three *GLIS3*$^{-/-}$ clones (KO22, KO28 and KO29) and two wild type (WT) clones (WT2 and WT7) along with the parental HES3 line (WT

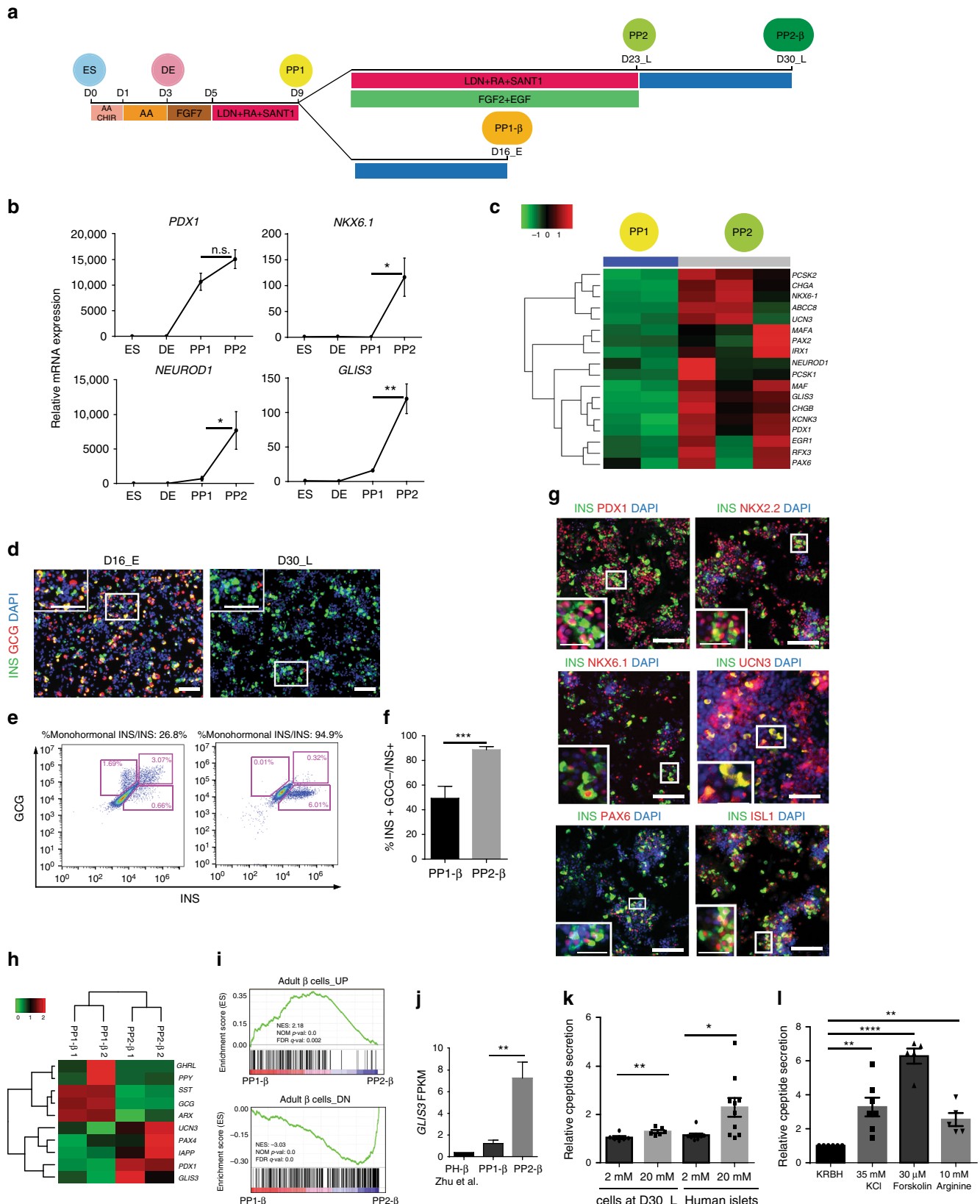

(PL)). Details for clonal lines used in each experiment are listed in Supplementary Table 7.

These isogenic WT and mutant lines were differentiated to D30_L using our established strategy (Fig. 1a). No significant difference was detected between WT and mutant lines with respect to their capacities to differentiate toward definitive endoderm (SOX17$^+$OCT4$^-$, Supplementary Fig. 2d–f) or PP1 (PDX1$^+$/ SOX9$^+$, Supplementary Fig. 2g–i). This is in agreement with a previous study that found no obvious defects in GLIS3$^{-/-}$ HUES8 cells[20]. At D23_L, no significant difference is detected between WT and mutant lines regarding the percentage of PDX1$^+$ cells (Supplementary Fig. 2k, l). However, qRT-PCR assays revealed decreased relative expression levels in GLIS3$^{-/-}$ PP2 cells for key pancreatic endocrine markers, including PDX1, NEUROD1, NKX6.1, and MAFA (Fig. 2a). RNA profiling (Fig. 2b) and GSEA (Fig. 2c, Supplementary Fig. 3a) showed a marked decrease in the expression of genes related to the pancreatic endocrine compartment in GLIS3$^{-/-}$ PP2 cells. Consistent with these data, gene ontology (GO, Fig. 2d) and KEGG pathway (Fig. 2e) analyses highlight downregulation of Type B pancreatic cell differentiation and endocrine pancreas development in GLIS3$^{-/-}$ PP2 cells. Taken together, these findings suggest that endocrine development is compromised in GLIS3$^{-/-}$ PP2 cells. To decipher whether the decrease in endocrine-related genes is due to decreased number of endocrine cells or a reduction of transcripts per cell, the number and percentage of endocrine progenitors (NGN3$^+$) and their derivatives (CHGA$^+$) were quantified at different time points during transition from D9 to D23_L (Supplementary Fig. 3b–f). There was no significant difference between WT and GLIS3$^{-/-}$ cells at any of the steps tested. However, we observed a decrease in fluorescence intensity of NGN3 staining in GLIS3$^{-/-}$ cells at D23_L, suggesting that loss of GLIS3 might decrease the expression of NGN3 per cell, instead of affecting the percentage of NGN3$^+$ cells (Supplementary Fig. 3g–i).

To determine whether loss of GLIS3 affects endocrine differentiation capacity, isogenic WT and GLIS3$^{-/-}$ hESCs were differentiated to D30_L and monitored for the expression of endocrine hormones. First, flow cytometry was used to quantify the percentage of INS-GFP$^+$ cells. A significant decrease of INS$^+$ cells was measured in the GLIS3$^{-/-}$ cells at D30_L (Fig. 2f, g). Intracellular flow cytometry and immunocytochemistry analyses for INS further validated the decrease of INS$^+$ cells in the GLIS3$^{-/-}$ cells at D30_L (Fig. 2h–k). In contrast, both percentage and absolute number of GHRL$^+$ cells (Supplementary Fig. 4a, b, i) and SST$^+$ cells were significantly increased in the GLIS3$^{-/-}$ cells at D30_L (Supplementary Fig. 4c, d, j). There was no significant change in GCG$^+$ cells (Supplementary Fig. 4e, f). The percentage of total endocrine-like cells (GCG$^+$ + INS$^+$ + SST$^+$ + GHRL$^+$) was not significantly different between WT and GLIS3$^{-/-}$ cells at

D30_L (Fig. 2i). Similar to earlier stages, there was no significant difference between WT and GLIS3$^{-/-}$ cells at D30_L regarding the percentage of CHGA$^+$ cells (Supplementary Fig. 4g, h). However, the ratio of defined subtypes of endocrine cells changes in the GLIS3$^{-/-}$ population, which compared to WT is comprised of more GHRL$^+$ ε-like cells and SST$^+$ δ-like cells at the expense of INS$^+$ β-like cells, while the GCG$^+$ α-like cells remain unchanged (Fig. 2j). This change of endocrine cell subtype in the GLIS3$^{-/-}$ population was further confirmed by immunocytochemistry analysis (Fig. 2k). Finally, we monitored the function of the derived GLIS3$^{-/-}$ at D30_L and found they respond in a similar manner to their WT counterparts when stimulated with 20 mM glucose and other secretagogues including KCl, Forskolin, and Arginine (Supplementary Fig. 4l, m), suggesting that the absence of GLIS3 does not affect glucose sensing or the insulin secretory machinery. However, the median fluorescence intensity of INS staining was significantly decreased in GLIS3$^{-/-}$ INS-GFP$^+$ PP2-β cells compared to WT INS-GFP$^+$ PP2-β cells (Fig. 2l, m). Consistently, ELISA using lysates of the purified WT or GLIS3$^{-/-}$ INS-GFP$^+$ PP2-β cells showed significantly lower c-peptide content in GLIS3$^{-/-}$ PP2-β cells (Fig. 2n). This phenomenon could be the result of lower INS transcription, as GLIS3 is known to bind to the INS promoter and activate its expression[30,31]. Indeed, FACS-purified GLIS3$^{-/-}$ INS-GFP$^+$ PP2-β cells had lower expression of the INS mRNA compared to WT cells (Supplementary Fig. 4k)

**Loss of GLIS3 leads to increased apoptosis.** Previous studies showed that knockdown of Glis3 induces apoptosis in a rat β-cell line[32]. Therefore, we assessed the viability of GLIS3$^{-/-}$ hESC-derived cells at different stages throughout the differentiation process. We monitored apoptosis of WT and GLIS3$^{-/-}$ hESCs, DE, PP1, and PP2 cells using Annexin V staining (Fig. 3a). The apoptotic rates of GLIS3$^{-/-}$ hESCs or DE cells are comparable to WT cells. At D9, there is a modest yet statistically significant higher apoptosis rate in GLIS3$^{-/-}$ PP1 cells compared to WT PP1 cells (8.2% ± 0.3 vs. 4.5% ± 0.7). The apoptosis rate is exacerbated in GLIS3$^{-/-}$ PP2 cells at D23_L. Around 21.9% ± 2.5 of GLIS3$^{-/-}$ PP2 cells stained positive for Annexin V, which is significantly higher than for WT PP2 cells (5.5% ± 0.4, Fig. 3a–c). To interrogate whether the increased apoptosis was a consequence of overcrowding, we measured proliferation rate of PP2 cells. There was no significant difference in the confluency or proliferation rate between GLIS3$^{-/-}$ and WT PP2 cells (Supplementary Fig. 5a, b). The percentage of Annexin V$^+$ cells and intensity of annexin V staining are likewise significantly higher in the GLIS3$^{-/-}$ INS-GFP$^+$ PP2-β cells compared to WT INS-GFP$^+$ PP2-β cells (GLIS3$^{-/-}$: 22.7 ± 2.0% vs. WT: 13.9 ± 2.1%,

**Fig. 1** Generation of mono-hormonal pancreatic β-like cells through the induction of late-stage pancreatic progenitors (PP2). **a** Schematic representation of the stepwise differentiation protocol. **b** qRT-PCR analysis of pancreatic progenitor markers in hESCs, definitive endoderm (DE), PP1 and PP2 cells (ES, DE n = 3, PP1 n = 6, PP2 n = 8). **c** Heatmap representing relative expression profiles of genes related to β-cell development in PP1 and PP2 cells (PP1 n = 2, PP2 n = 3). **d** Immunocytochemistry analysis of insulin (INS) and glucagon (GCG) expression at D16_E and D30_L, insets show a higher magnification image. Scale bar = 100 μm. Intracellular flow cytometry analysis (**e**) and quantification (**f**) of INS and GCG expression at D16_E and D30_L (D16_E n = 4, D30_L n = 8). **g** Immunocytochemistry analysis of cells at D30_L. Scale bar = 100 μm, scale bar of high magnification insets = 40 μm. **h** Heatmap representing the relative expression level of endocrine markers in PP1-β and PP2-β cells. **i** GSEA analysis shows that PP2-β cells are transcriptionally closer to human adult β-cells than PP1-β cells. The adult β cells_UP and adult β cells_DN gene sets consist of the top 1000 differentially expressed genes (higher expression for UP and lower expression for DN) in primary human β-cells compared to PP1-β cells. **j** FPKM values of GLIS3 in PH-β cells derived using Zhu et al.'s protocol and the purified INS-GFP$^+$ PP1-β (n = 4) and PP2-β cells (n = 2). **k** Glucose-stimulated c-peptide secretion of cells at D30_L and human islets. The amount of c-peptide secretion in 20 mM (high) D-glucose condition was normalized to the amount of c-peptide secreted in 2 mM (low) D-glucose condition (n = 6 for cells at D30_L and n = 10 for islets). **l** Insulin secretion of cells at D30_L in response to other secretagogues, including 30 mM KCl (n = 7), 30 μM Forskolin (n = 5), or 10 mM Arginine (n = 5) relative to basal Krebs–Ringer bicarbonate HEPES (KRBH) buffer treatment (n = 7). The fold change was normalized to the amount of c-peptide secreted in KRBH condition. P values by unpaired two-tailed t-test were *P < 0.05, **P < 0.01, ***P < 0.001, ****P < 0.0001. Data are presented as individual biological replicates. The center value is "mean". Error bar is SEM

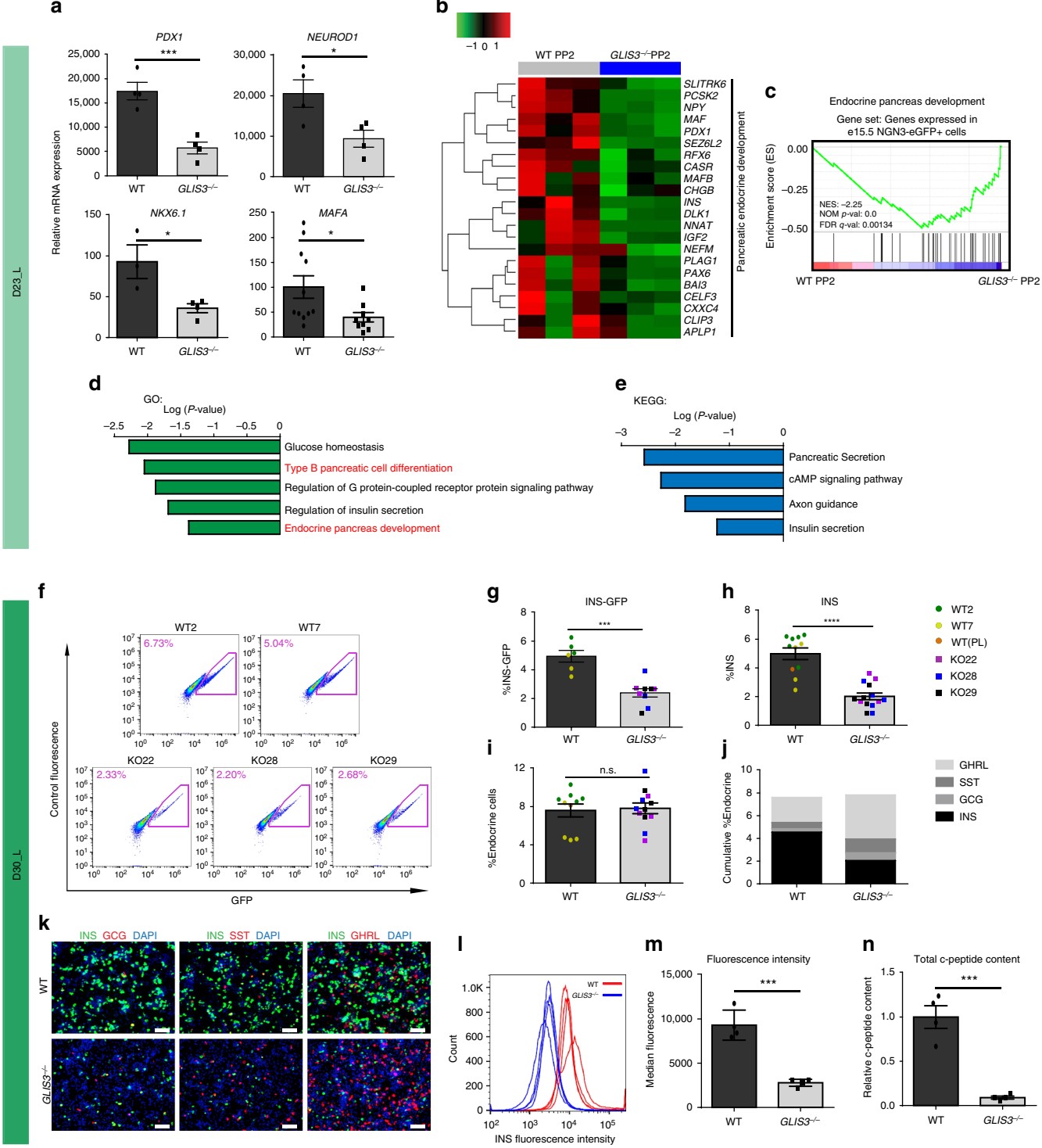

**Fig. 2** Biallelic mutation of *GLIS3* affects pancreatic differentiation and the generation of endocrine cells. **a** qRT-PCR analysis of pancreatic markers in the isogenic WT and *GLIS3*[−/−] PP2 cells. (*PDX1, NEUROD1, NKX6.1 n = 4, MAFA n = 12*). **b** Heatmap representing the relative expression levels of endocrine genes in WT and *GLIS3*[−/−] PP2 cells (*n = 3*). **c** GSEA analysis showing the decrease of endocrine pancreas-related genes in *GLIS3*[−/−] PP2 cells. Gene ontology (GO) analysis (**d**) and KEGG pathway analysis (**e**) of genes significantly downregulated (*P < 0.01*) in *GLIS3*[−/−] PP2 cells. **f** Flow cytometry analysis of WT and *GLIS3*[−/−] cells at D30_L. **g** Quantification of the percentage of INS-GFP+ cells of WT (*n = 6*) and *GLIS3*[−/−] (*n = 9*) cells at D30_L. The percentage of INS-GFP+ cells was quantified based on flow cytometry analysis of GFP+ cells. **h** Quantification of the percentage of INS+ cells of WT (*n = 11*) and *GLIS3*[−/−] (*n = 14*) cells at D30_L. The percentage was quantified based on intracellular flow cytometry analysis of INS+ cells. **i** Total percentage of endocrine cells in isogenic WT (*n = 10*) and *GLIS3*[−/−] (*n = 12*) cells at D30_L. The percentage of endocrine cells is calculated as the sum of the percentages of INS+, GCG+, SST+, and GHRL+ cells. **j** Plot representing the ratios of different endocrine subtypes in WT and *GLIS3*[−/−] cells at D30_L. **k** Immunocytochemistry analysis of pancreatic endocrine marker expression in WT and *GLIS3*[−/−] cells at D30_L. Scale bar = 100 μm. **l, m** Histogram showing fluorescence intensity (**l**) and quantification of median fluorescence values (**m**) of INS staining of WT and *GLIS3*[−/−] PP2-β cells (*n = 4*). **n** Insulin content of the purified INS-GFP+ WT and *GLIS3*[−/−] PP2-β cells (*n = 4*). Data are normalized to the WT mean value. *P* values by unpaired two-tailed *t*-test were *P < 0.05, **P < 0.01, ***P < 0.001, ****P < 0.0001. The center value is "mean". Error bar is SEM

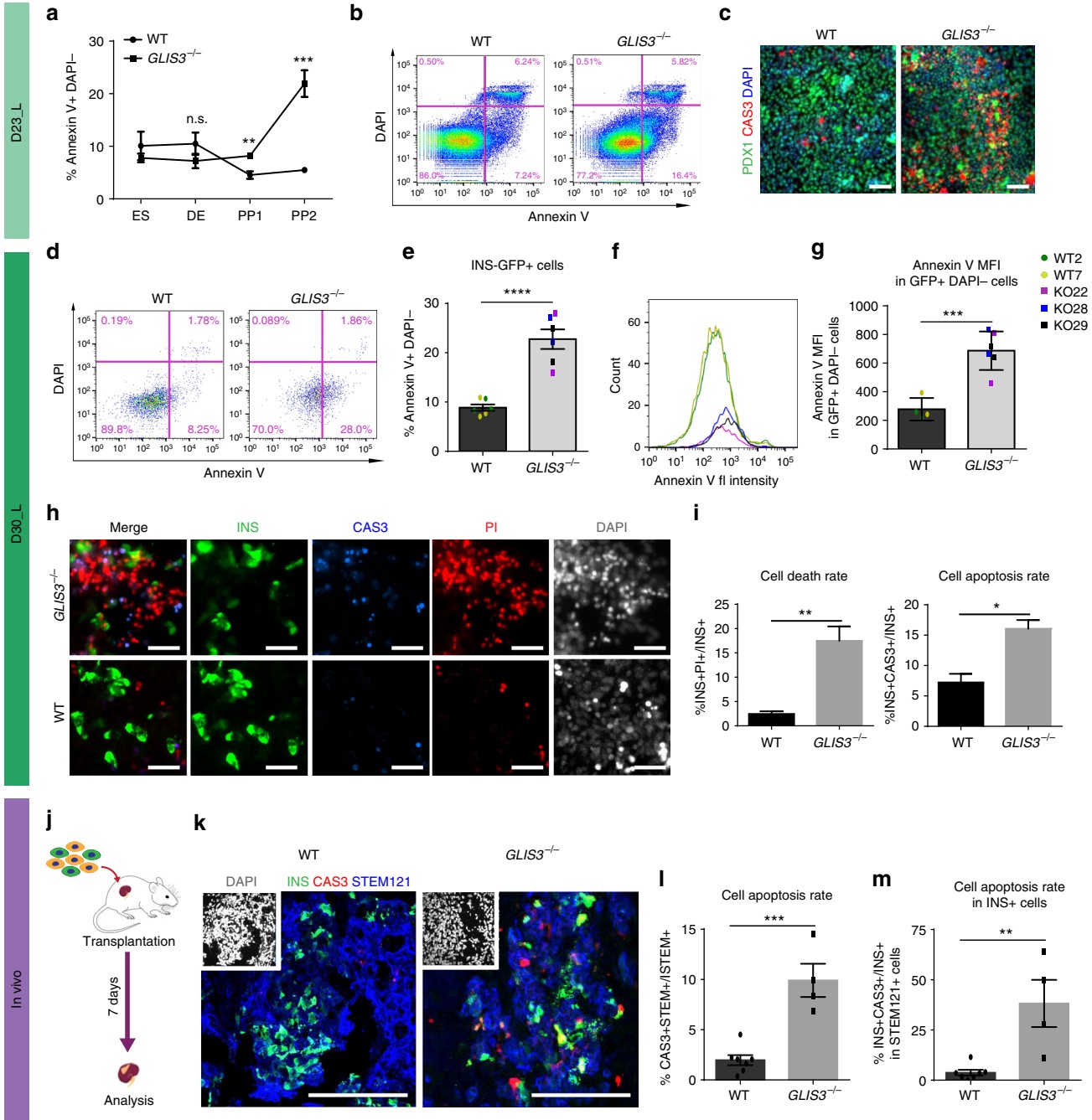

**Fig. 3** Loss of *GLIS3* leads to increased cell death in PP2 and PP2-β cells. **a** Quantification of early apoptotic cells (the percentage of Annexin V+/DAPI− cells) in WT and *GLIS3*−/− ES, DE, PP1, and PP2 cells (*n* = 3). **b** Representative flow cytometry analysis plots of Annexin V staining in WT and *GLIS3*−/− cells at D23_L. **c** Immunostaining for PDX1 and cleaved caspase-3 in WT and *GLIS3*−/− cells at D23_L. Scale bar = 40 μm. Annexin V staining (**d**) and quantification (**e**) of early apoptotic cells in the INS-GFP+ cells at D30_L (*n* = 6). Histogram showing fluorescence intensity (**f**) and quantification of median fluorescence values (**g**) of Annexin V staining of WT and *GLIS3*−/− INS+ DAPI−PP2-β cells (WT *n* = 4, *GLIS3*−/− *n* = 6). **h** PI, cleaved caspase-3 and INS staining of WT and *GLIS3*−/− cells at D30_L. Scale bar = 40 μm. **i** Quantification of cell death rate (the percentage of PI+INS+ cells in INS+ cells) and apoptosis rate (the percentage of cleaved caspase-3+INS+ cells in INS+ cells) of WT and *GLIS3*−/− INS+ PP2-β cells (*n* = 3). **j** Schematic representation of the in vivo transplantation experiment. **k** Immunostaining for INS, cleaved caspase-3 and STEM121 in the grafts of mice transplanted with WT or *GLIS3*−/− cells. Scale bar = 100 μm. **l** Quantification of the apoptosis rate (the percentage of cleaved caspase-3+/ STEM121+ cells in STEM121+ cells) within WT and *GLIS3*−/− grafts (*n* = 7 for WT, *n* = 4 for *GLIS3*−/−). **m** Quantification of the percentage of apoptotic INS+ cells (CAS3+PDX1+ STEM121+) in the INS+ population within the WT and *GLIS3*−/− grafts (INS+STEM121+, WT *n* = 7, *GLIS3*−/− *n* = 4). CAS3: cleaved caspase-3. *P* values by unpaired two-tailed Student's *t*-test were *\**P* < 0.05, \*\**P* < 0.01, \*\*\**P* < 0.001, \*\*\*\**P* < 0.0001. The center value is "mean". Error bar is SEM

Fig. 3d–g). Immunocytochemistry analysis using antibodies to stain INS, cleaved caspase-3 in addition to PI further confirmed the higher cell death rate (the percentage of PI[+]INS[+] cells in INS[+] cells) and cell apoptosis rate (the percentage of cleaved caspase-3[+] INS[+] cells in INS[+] cells, Fig. 3h, i) in GLIS3[−/−] INS-GFP[+] PP2-β cells than those in WT INS-GFP[+] PP2-β cells. Similar to GLIS3[−/−] INS-GFP[+] PP2-β cells, the GLIS3[−/−] INS-GFP[−] cells at D30_L show higher cell death and apoptosis compared to WT INS-GFP[−]cells (Supplementary Fig. 5c–g); the majority of the INS[−] cells at D30_L are undifferentiated PDX1[+] cells (Supplementary Fig. 5e). Intriguingly, the decrease of cellular viability in the GLIS3[−/−] cells correlates closely with expression of GLIS3 during the differentiation process (Fig. 1b), suggesting an essential role for GLIS3 for the survival of PP2 and PP2-β cells. To determine whether loss of GLIS3 leads to increased apoptosis of cells in vivo, WT or GLIS3[−/−] cells at D23_L were transplanted under the kidney capsule of 6–8-week-old male SCID-beige mice (Fig. 3j). The mice were sacrificed after 7 days and the grafts were examined for apoptosis by staining for cleaved caspase-3. Human cells were identified by labeling for human cytoplasmic marker STEM121. Consistent with our in vitro results (Fig. 3a–i), a significantly increased cell apoptosis rate (the percentage of cleaved caspase-3[+]/ STEM121[+] cells in STEM121[+] cells, Fig. 3k, l) was detected in the grafts of mice transplanted with GLIS3[−/−] cells. Furthermore, the cell apoptotic rate in INS[+] and PDX1[+] cells of the GLIS3[−/−] grafts is higher than for WT grafts. (Fig. 3m and Supplementary Fig. 5h, i).

**Galunisertib rescues apoptosis induced by loss of GLIS3**. Loss of function mutations in GLIS3 cause neonatal diabetes[8] and increased cell death caused by GLIS3 mutations may also contribute to T1D and T2D. Having access to the disease-relevant cells presenting a clear disease phenotype, we carried out a high-content chemical screen to identify drug candidates that can rescue the increased cell death in GLIS3[−/−] cells. First, we performed a time course experiment to optimize the time window for the chemical screen and determined that cell apoptosis significantly increases from day 25 to day 29 of the differentiation protocol (Supplementary Fig. 6a). Thus, the screen was carried out from day 25 to day 29 to identify drug candidates that block cell apoptosis induced by loss of GLIS3. To perform the screen, GLIS3[−/−] cells at D25_L were replated in 384-well plates and then exposed to compounds from multiple chemical libraries, including FDA approved drugs and drugs in clinical trials, kinase inhibitors, signaling pathway modulators, and other annotated compounds. These compounds were added at 1 or 10 μM for 4 days followed by high-content microscopy and analysis to determine the number of apoptotic and INS[+] cells post exposure (Fig. 4a and Supplementary Fig. 6b). In the DMSO-treated condition, the percentage of cleaved caspase-3[+] cells was 12.6 ± 0.9%. Compounds that decreased the percentage of cleaved caspase-3[+] cells by at least 2.5-fold (z score ≤ −1.5) were picked as primary hits. After screening ~5000 compounds, we identified 23 primary hit compounds that prevented the increased cell death in GLIS3[−/−] INS[−]GFP[+] PP2-β cells (Supplementary Fig. 6c and Supplementary Table 8). The Z′ factor of the primary screen was 0.52. The signal to basal ratio was 0.2 with a coefficient of variation 0.39. The hit rate was ~0.45%. After several rounds of hit confirmation, we focused on one compound, galunisertib (LY2157299), that showed the highest efficacy and replicability (Fig. 4b). Galunisertib efficiently rescues loss of GLIS3-induced pancreatic β-cell apoptosis in a dose-dependent manner (IC50: 1.14 μM, Fig. 4c). Treatment of cells with 10 μM galunisertib significantly reduces cell death (Fig. 4d, e and Supplementary Fig. 8a) and cell apoptosis (Fig. 4d, f–h and Supplementary Fig. 8b) in GLIS3[−/−] cells at D30_L.

Consistently, galunisertib treatment increases the number of INS[+] cells (Fig. 4i). We also tested galunisertib using WT INS-GFP[+] PP2-β cells. Galunisertib does not affect cell apoptosis of WT INS[+] cells, showing that the rescue effect of galunisertib is specific to GLIS3[−/−] cells (Supplementary Fig. 7a, b).

We further tested galunisertib in vivo. GLIS3[−/−] PP2 cells were pre-treated with 10 μM galunisertib for 16 h and then transplanted under the kidney capsule of 6–8-week-old male SCID-beige mice (Fig. 4j). The mice were randomly separated into two groups and treated with either 15 mg kg[−1] day[−1] of galunisertib or vehicle by intraperitoneal injection for 7 days. The mice were euthanized and the grafts were analyzed for cell apoptosis. Galunisertib treatment significantly decreases the cell apoptosis rate in the transplanted cells (the percentage of cleaved caspase-3[+]/ STEM121[+] cells in STEM121[+] cells, Fig. 4k, l) and more specifically in INS[+] and PDX1[+] cells in the grafts (Fig. 4m and Supplementary Fig. 8c, d), showing that in vivo galunisertib can rescue cell death induced by loss of GLIS3.

**Loss of GLIS3 causes cell death by activating TGFβ pathway.** Galunisertib was previously developed as a TGFβR 1 kinase inhibitor[33,34]. To investigate the mechanism of action, we used RNA-seq profiling to compare the WT and GLIS3[−/−] PP2 cells. KEGG pathway analysis (Fig. 5a) highlights the upregulation of the TGFβ signaling pathway in GLIS3[−/−] PP2 cells while GO analysis (Fig. 5b) further validates the upregulation of positive regulation of pathway restricted SMAD protein phosphorylation and SMAD protein signal transduction in GLIS3[−/−] PP2 cells. We observed significant upregulation in key genes involved in TGFβ activation including TGFB2, TGFB3, and TGFBR2 (Supplementary Fig. 9a). Together, these data point to an upregulation of the TGFβ pathway in GLIS3[−/−] PP2 cells. Western blotting confirmed an increased SMAD2/3 phosphorylation level in GLIS3[−/−] PP2 cells (Fig. 5c, d and Supplementary Fig. 10). Importantly, pSMAD2/3 staining co-localizes with INS in GLIS3[−/−] PP2-β cells (Supplementary Fig. 9b). RNA-seq profiling of FACS-purified INS-GFP[+] WT and GLIS3[−/−] PP2-β cells confirmed the relative upregulation of TGFβ pathway-related genes in the GLIS3[−/−] PP2-β cells (KEGG pathway analysis, P-value < 0.05, Fig. 5e). Ingenuity Pathway Analysis (IPA) predicts upregulation of the TGFβ pathway in GLIS3[−/−] PP2-β cells (Supplementary Fig. 9c). Galunisertib inhibits the increased SMAD2/3 phosphorylation in GLIS3[−/−] cells at D30_L (Fig. 5f, g and Supplementary Fig. 10). Finally, we tested other TGFβ inhibitors. Consistent with galunisertib, a range of TGFβ inhibitors, including 1 μM SB431542, 10 μM A83–01, 1 μM SB525334, and 10 μM LY-364947, significantly decrease the cell death and cell apoptosis rates in INS-GFP[+] GLIS3[−/−] PP2-β cells (Fig. 5h, i and Supplementary Table 9). These TGFβ inhibitors also prevent increased cell apoptosis and cell death in INS[−] GLIS3[−/−] cells at D30_L, which are mainly undifferentiated PDX1[+] cells (Supplementary Fig. 9d, e). Collectively, these findings indicate that galunisertib prevents increased cell death by inhibiting an inappropriate upregulation of the TGFβ signaling pathway in GLIS3[−/−] cells.

**Discussion**
Significant progress has been made directing the differentiation of hPSCs into functional β-like cells for regenerative medicine[27–29]. Considering the ultimate goals of these studies are to derive enough cells for cell replacement therapy, these protocols use a variety of growth factors and small molecules to maximize the β-cell yield. Such conditions, however, may not be optimal for disease modeling since the modulation of many pathways in these protocols may outweigh the developmental competence of the differentiating progenitors and mask subtle differences between

healthy and diseased cells. Thus, there is a strong rationale to develop "minimal component" protocols for disease modeling. Here, we describe a stepwise protocol that closely mimics pancreatic development, through generation of GLIS3-expressing late-stage pancreatic progenitors that generate mono-hormonal glucose-responding β-like cells. The GSIS response of cells at D30_L is not indistinguishable from human primary islets.

Additional optimization might be required to further increase the response of PP2-β cells to glucose stimulation. These late-stage progenitors resemble second transition cells in mouse development. Our strategy incorporates limited manipulation of developmental pathways and serves thus far as an optimized protocol to model the pancreatic β-cell defects in human diabetes.

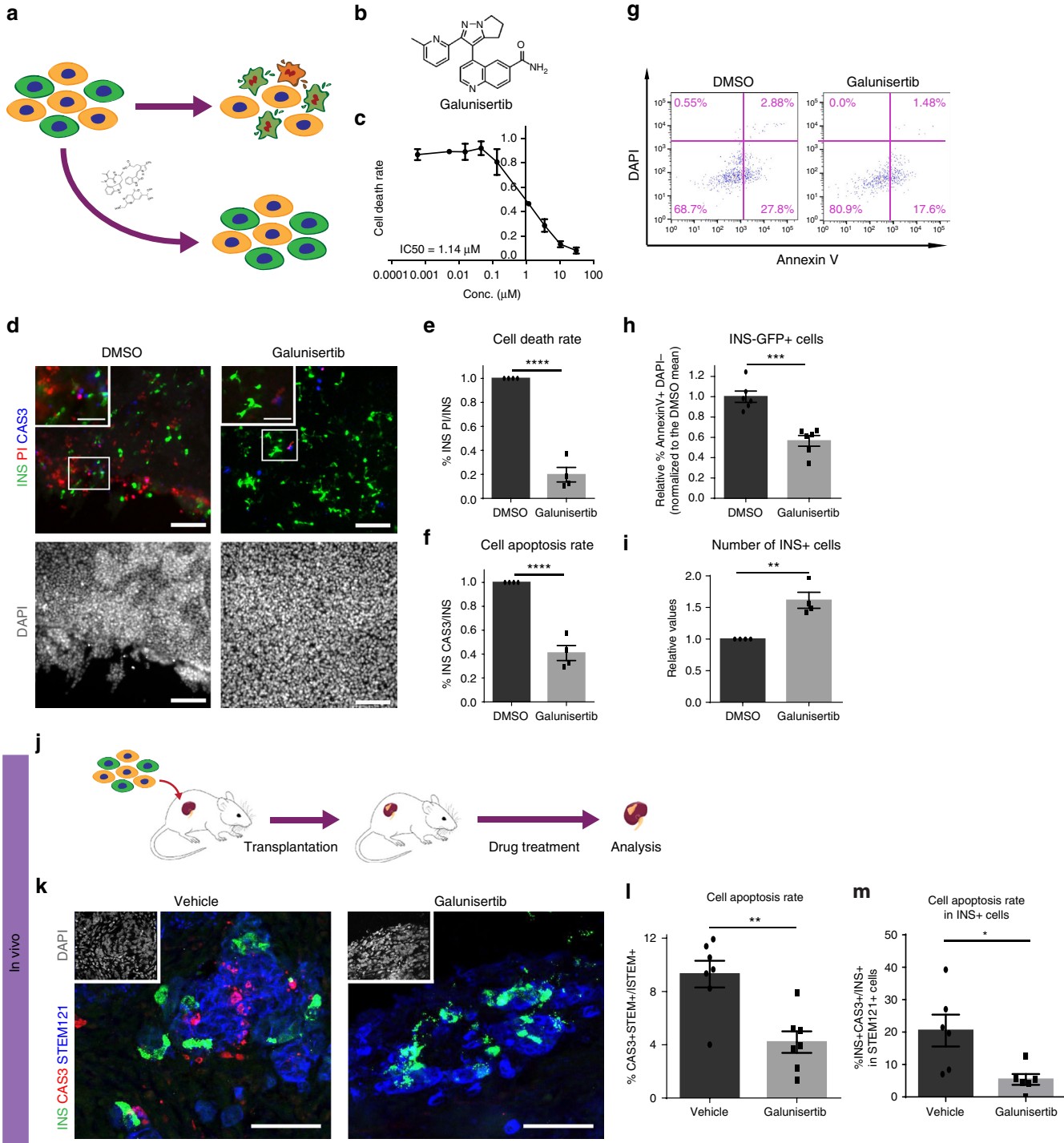

Using this "minimal component" differentiation strategy, we studied the role of *GLIS3* in different stages of pancreatic differentiation. Our initial attempt using isogenic *GLIS3*$^{-/-}$ hESCs failed to recapitulate the defects observed in *Glis3*$^{-/-}$ mice[20], which might be due to lack of GLIS3 expressing cells. It should be noted that we cannot fully exclude the possibility that the previous mutations were not total loss-of-function. Using our newly developed "minimal component" differentiation strategy, we found that loss of *GLIS3* does not affect the induction of DE and PP1 cells, in agreement with an earlier study[20] and consistent with findings that the number of pancreatic progenitors in *Glis3*$^{-/-}$ mice is not significantly different from WT mice[12]. By quantification of different endocrine subtypes, we found that the percentage of INS$^+$ cells is significantly decreased in *GLIS3*$^{-/-}$ cells, also seen in *Glis3*$^{-/-}$ mice[12]. However, in our model, the total size of the human endocrine cell pool is not significantly affected, and the proportion of GHRL$^+$ ε-like and SST$^+$ δ-like cells increases at the expense of INS$^+$ β-like cells in the *GLIS3*$^{-/-}$ hESC-derived population. This suggests that loss of *GLIS3* directs the differentiation of endocrine progenitors along the GHRL$^+$ ε-like and SST$^+$ δ-like cells instead of INS$^+$ β-like cells lineage. In contrast, the number of endocrine cells including all subtypes is markedly reduced in *Glis3*$^{-/-}$ mice. The difference between *GLIS3*$^{-/-}$ hESC-derived cells and *Glis3*$^{-/-}$ mice might underline the distinction between mouse development and human cell-based systems or the difference between in vitro or in vivo conditions. Yet, it emphasizes the importance of including human models to study diabetes-associated genes.

Previous studies have suggested that knockdown of *Glis3* induces apoptosis[32]. However, the downstream pathways regulated by GLIS3 are unknown. We documented remarkably increased rates of cell apoptosis and cell death in *GLIS3*$^{-/-}$ hESC-derived PP2 and PP2-β cells, which is consistent with earlier findings on the role of Glis3 in β-cell viability in rodents. RNA-seq analysis suggested upregulation of TGFβ signaling in *GLIS3*$^{-/-}$ PP2 and PP2-β cells, which was further confirmed at the protein level by western blotting. Activated TGFβ signaling has been documented to play important roles in inducing cellular stress and apoptosis in many different cell types. In accordance with these data, treatment of *GLIS3*$^{-/-}$ PP2-β cells with TGFβ inhibitors rescues cell death induced in the absence of GLIS3.

An important goal of developing isogenic hESCs carrying disease-associated mutations is to create a robust and disease-relevant platform for drug discovery. We used *GLIS3*$^{-/-}$ hESC-derived β-like cells to perform a screen using FDA approved drugs and drugs in clinical trials. We discovered that galunisertib, a drug candidate in Phase II clinical trial[35], can effectively rescue cell death in *GLIS3*$^{-/-}$ β-like cells both in vitro and in vivo. More importantly, galunisertib does not affect WT cells, suggesting that the effect of galunisertib is specific to *GLIS3*$^{-/-}$ β-like cells. Previous studies have shown that loss of *GLIS3* function results in neonatal diabetes[8]. While administration of an inhibitor of TGFβ signaling during pregnancy could impose complications, it is possible that it could have potential for GLIS3-associated diabetes in the adults. Studies in mice show that normal levels of Glis3 are required for β-cell homeostasis during adulthood. β-cell specific inactivation of *Glis3* leads to β-cell loss and diabetes. Additionally, *Glis3*$^{+/-}$ mice are predisposed to diabetes and become diabetic under high-fat diet[15]. *GLIS3* SNPs are strongly associated with both T1D[1,3] and T2D[5–7]. It would be intriguing to evaluate whether these variants lead to lower GLIS3 levels leading to perturbations of TGFβ signaling inside β-cells which can be prevented by galunisertib administration. Thus, our study has wide-ranging implications for the treatment of neonatal diabetes and broadens the scope of precision medicine for more complex conditions, including T1D and T2D.

## Methods

**Maintenance of hESCs.** Human ESC lines INS$^{GFP/W}$ HES3, HUES8, and H1 were grown on Matrigel-coated 10 cm$^2$ plates in mTeSR1 medium (STEMCELL Technologies) supplemented with 50 μg mL$^{-1}$ Normocin (InvivoGen). Cells were maintained at 37 °C with 5% CO$_2$. Cultures were passaged every 4–6 days at 1:15–1:20 with 0.5 mM EDTA. All lines were routinely tested for mycoplasma contamination. All hESC studies were approved by the Tri-Institutional Embryonic Stem Cell Research Committee (ESCRO).

**In vitro differentiation of hESCs.** To prepare for differentiation, hESCs were dissociated with 0.5 mM EDTA and plated on Matrigel-coated 6-well plates at a ratio of 1:1–1:2 resulting at ~95% starting confluency. The differentiation started 24–48 h later. On day 0, cells were exposed to basal medium RPMI 1640 supplemented with 1× Glutamax (ThermoFisher Scientific), 50 μg mL$^{-1}$ Normocin, 100 ng mL$^{-1}$ Activin A (R&D), and 2 μM of CHIR99021 (GSK3β inhibitor 3, SelleckChem) for 24 h. The medium was changed on day 1 to basal RPMI 1640 supplemented with 1× Glutamax (ThermoFisher Scientific), 50 μg mL$^{-1}$ Normocin, 0.2% fetal bovine serum (Corning), 100 ng mL$^{-1}$ Activin A (R&D) for 2 days. On day 3, the resulting definitive endoderm cells were cultured in basal RPMI 1640 supplemented with 1× Glutamax (ThermoFisher Scientific), 50 μg mL$^{-1}$ Normocin, 2% fetal bovine serum (Corning), 50 ng mL$^{-1}$ FGF7 (Peprotech) for 2 days to acquire foregut fate. On day 5, the cells were induced to differentiate into pancreatic endoderm in basal medium DMEM 4.5 g L$^{-1}$ glucose (Corning) supplemented with 1× Glutamax, 50 μg mL$^{-1}$ Normocin and 2% B27 (GIBCO), 2 μM retinoic acid (RA; Sigma), 200 nM LDN193189 (LDN, Stemgent), and 0.25 μM SANT-1 for 4 days (PP1). The medium was subsequently refreshed every other day. On day 9, this medium was further supplemented with 10 ng mL$^{-1}$ EGF (Peprotech) and 10 ng mL$^{-1}$ FGF2 to help maintain the cells at the pancreatic progenitor stage. In the case of the H1 line, cells were treated with 3 μM RA, 200 nM LDN, 0.25 μM SANT-1, 15 ng mL$^{-1}$ EGF, and 15 ng mL$^{-1}$ FGF2 for a 14-day period. On day 23, the PP2 cells differentiate into late-stage INS$^+$ PP2-β-like cells in basal differentiation medium including DMEM supplemented with 1× Glutamax, 50 μg mL$^{-1}$ Normocin, 2% B27 for 7 days (D30_L). For differentiation to PP1-β cells, PP1 cells on day 9 were cultured for 7 days in the basal differentiation medium (D16_E).

**Generation of isogenic GLIS3 mutant lines.** To mutate the human *GLIS3* gene, two sgRNAs targeting exon three of the gene were designed and cloned into a vector carrying a CRISPR-Cas9 gene (Addgene plasmid #42230). The sgRNAs were validated using the surveyor assay in 293T cells. The construct containing validated sgRNA was then co-electroporated together with a vector expressing puromycin into dissociated INS$^{GFP/W}$ HES3 cells suspended in Human Stem Cell Nucleofector solution (Lonza) following the manufacturer's instructions. After replating, the electroporated cells were selected with 500 ng mL$^{-1}$ puromycin. After 2 days of puromycin selection, hESCs were dissociated into single cells by Accutase (Innovative Cell Technologies) and replated at low density. The cells were supplemented

**Fig. 4** A high-content chemical screen identifies galunisertib as a drug candidate to rescue cell death induced by loss of GLIS3 both in vitro and in vivo. **a** Schematic representation of the high-content chemical screen. **b** Chemical structure of galunisertib. **c** Inhibitory curve of galunisertib. **d** Immunocytochemistry analysis of *GLIS3*$^{-/-}$ PP2-β cells treated with DMSO or 10 μM galunisertib. Scale bar = 100 μm, scale bar of high magnification insets = 40 μm. **e, f** Quantification of the cell death rate (**e**, the percentage of PI$^+$INS$^+$ cells in INS$^+$ cells, $n = 4$) and apoptosis rate (**f**, the percentage of cleaved caspase-3$^+$INS$^+$ cells in INS$^+$ cells, $n = 3$) of *GLIS3*$^{-/-}$ PP2-β cells treated with DMSO or 10 μM galunisertib. Flow cytometry analysis (**g**) and quantification (**h**) of early apoptotic cells (the percentage of Annexin V$^+$/DAPI$^-$ cells) in *GLIS3*$^{-/-}$ INS-GFP$^+$ PP2-β cells treated with DMSO or 10 μM galunisertib ($n = 6$). **i** Relative number of INS$^+$ cells in *GLIS3*$^{-/-}$ PP2-β cells treated with DMSO or 10 μM galunisertib. Data are normalized to DMSO-treated values ($n = 4$). **j** Schematic representation of the in vivo transplantation and drug treatment experiments. **k** Immunohistochemistry analysis of INS, cleaved caspase-3, and STEM121 in the grafts isolated from vehicle- or galunisertib-treated mice. Scale bar = 100 μm. **l** Quantification of immunohistochemistry data in **j** ($n = 7$). **m** Quantification of the percentage of apoptotic INS$^+$ cells (CAS3$^+$INS$^+$ STEM121$^+$) in the INS$^+$ population within the grafts from vehicle- or galunisertib-treated mice (INS$^+$STEM121$^+$, $n = 6$). CAS3: cleaved caspase-3. $P$ values by unpaired two-tailed $t$-test were *$P < 0.05$, **$P < 0.01$, ***$P < 0.001$, ****$P < 0.0001$. The center value is "mean". Error bar is SEM

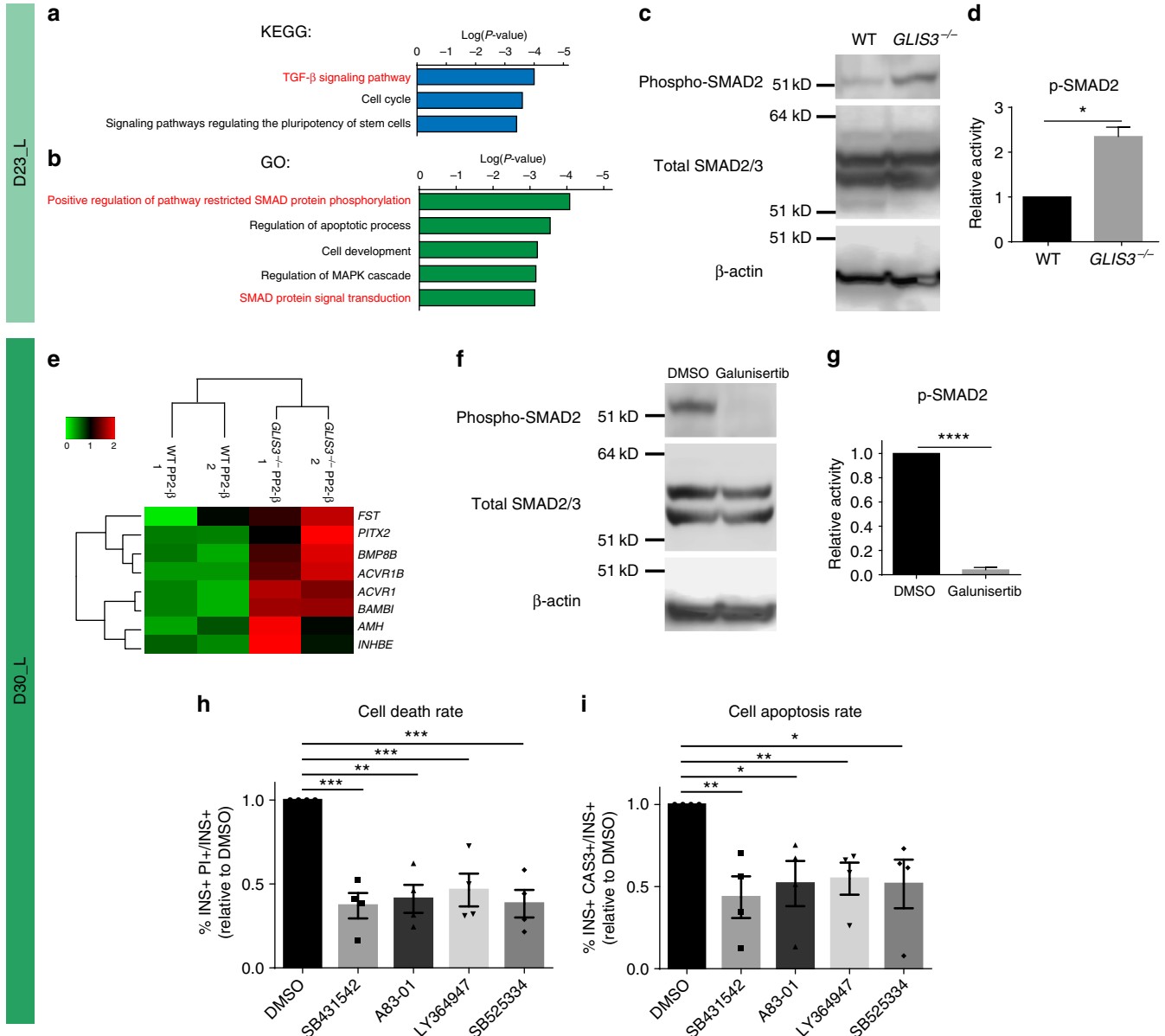

**Fig. 5** Galunisertib rescues loss of GLIS3 induced cell death by inhibiting TGFβ signaling. KEGG pathway analysis (**a**) and gene ontology (GO) analysis (**b**) of genes that are significantly ($P < 0.01$) upregulated in $GLIS3^{-/-}$ PP2 cells. Western blot analysis (**c**) and quantification (**d**, $n = 3$) of SMAD2/3 phosphorylation in WT and $GLIS3^{-/-}$ cells at D23_L. **e** Heatmap representing the relative expression levels of TGF-β-related genes upregulated at least two-fold in the purified WT and $GLIS3^{-/-}$ INS-GFP$^+$ PP2-β cells. Western blotting analysis (**f**) and quantification (**g**, $n = 3$) of SMAD2/3 phosphorylation in $GLIS3^{-/-}$ cells at D30_L treated with DMSO or 10 μM galunisertib. **h**, **i** Quantification of the cell death rate (**h**, the percentage of PI$^+$INS$^+$ cells in INS$^+$ cells) and apoptosis rate (**i**, the percentage of cleaved caspase-3$^+$INS$^+$ cells in INS$^+$ cells) of $GLIS3^{-/-}$ PP2-β cells treated with different TGFβ inhibitors ($n = 4$). CAS3: cleaved caspase-3. $P$ values by unpaired two-tailed $t$-test were *$P < 0.05$, **$P < 0.01$, ***$P < 0.001$, ****$P < 0.0001$. The center value is "mean". Error bar is SEM

with 10 μM Y-27632 for the first two days. After approximately 10 days, individual colonies were picked, mechanically disaggregated, and replated into two individual wells of 96-well plates. A portion of the cells was lysed and analyzed by Sanger sequencing. For biallelic frameshift mutants, we chose both homozygous mutants and compound heterozygous mutants. WT clonal lines from the targeting experiment were included as WT controls to account for potential non-specific effects associated with the gene-targeting process.

**Immunofluorescence staining**. Cells were fixed in 4% paraformaldehyde solution (Affymetrix) for 20 min, then blocked and permeabilized in PBS solution containing 5% horse serum and 0.3% Triton for 1 h at room temperature. The cells were incubated with primary antibodies overnight at 4 °C followed by 1 h incubation with fluorescence-conjugated secondary antibodies (Alexafluor, Thermo-Fisher Scientific) at RT. For pSMAD2/3 staining, cells were permeabilized with ice-cold methanol at −20 °C for 10 min after fixation and prior to blocking. The following primary antibodies were used: anti-OCT4 (1:200, Santa Cruz), anti-

SOX17 (1:500, R&D), anti-PDX1 (1:500, R&D), anti-SOX9 (1:1000, Millipore), anti-NKX6.1 (1:500, DSHB), anti-NKX2.2 (1:500, DSHB), anti-PAX6 (1:1000, Covance), anti-ISL1 (1:200, DSHB), anti-UCN3 (1:500, Pheonix Pharmaceuticals), anti-NGN3 (1:500, R&D), anti-chromogranin A (1:1000, Immunostar), anti-glucagon (1:2000, Sigma), anti-somatostatin (1:1000, DAKO), anti-ghrelin (1:500, Santa Cruz), anti-insulin (1:500, DAKO), and anti-cleaved caspase-3 (1:1000, BD Biosciences), anti-pSMAD2/3 (1:200, Cell Signaling).

**Flow cytometry and intracellular FACS analysis**. hESC-derived cells were dissociated using Accutase. To analyze GFP expression, the cells were resuspended in PBS and used directly for analysis. For intracellular staining, the cells were fixed and stained using Foxp3 staining buffer set (eBiosciences) according to the manufacturer's instructions. Briefly, cells were first blocked with 2% horse serum for 15 min and then incubated with primary antibody for 45 min at RT, washed twice, incubated with fluorescence-conjugated secondary antibody for 30 min at 4 °C, washed twice and re-suspended in FACS buffer for analysis. The following

primary antibodies were used: anti-SOX17 (1:500, R&D), anti-PDX1 (1:500, R&D), anti-pro-insulin (1:500, Millipore), anti-glucagon (1:100, Cell Signaling), anti-somatostatin (1:1000, DAKO), and anti-ghrelin (1:500, Santa Cruz). Samples were analyzed with an Accuri C6 flow cytometry instrument and the data were processed using Flowjo v10 software.

**Annexin V cellular apoptosis analysis.** hESC-derived cells were dissociated by Accutase and washed with cold PBS, stained with the PE/Annexin V apoptosis detection Kit (BD Bioscience, 559763) or A647-conjugated annexin V (Thermo Fisher Scientific) according to manufacturer's instructions, the samples were the analyzed by flow cytometry (BD Bioscience, FASC ARIA2) within 30 min. To include a positive control for apoptosis, cells were incubated with 10 µM Camptothecin (Sigma Aldrich) or DMSO for 4 h prior to Annexin V staining.

**Insulin secretion assays.** Cells were starved in 2 mL glucose-free DMEM (with GlutaMax) for 3 h followed by 1 h incubation in KRBH buffer (with 0.1% BSA) in a 5% $CO_2$/37 °C incubator. To perform GSIS, cells were exposed sequentially to 400 µL of KRBH, 2 mM glucose, and 20 mM glucose; supernatants were collected after 30 min and spun down to eliminate the cells and debris. The same procedure was carried out for treatments with 30 mM KCl, 10 mM arginine (Sigma A5006), or 30 µM forskolin. Supernatants were used for ELISA (Human C-peptide ELISA kit, Millipore, EZHCP-20K). To measure the total c-peptide levels in each sample, cells were lysed in RIPA buffer supplemented with 1× protease inhibitor cocktail (ThermoFisher Scientific) for 3 h at 4 °C. Lysates were spun down and supernatant was used for ELISA (Human C-peptide ELISA kit, Millipore, EZHCP-20K). C-peptide secretion from cells in each condition was normalized to KRBH treatment.

**Insulin content measurement.** Cells at D30_L were dissociated using Accutase and resuspended in DMEM containing 2% FBS and 1 mM EDTA; 20,000 INS-GFP$^+$ DAPI$^-$ cells were FACS sorted by an ARIA2 instrument, washed once with PBS and lysed in 200 µL RIPA buffer supplemented with 1× protease inhibitor cocktail (ThermoFisher Scientific). The Insulin content in the lysates was measured by ELISA (Human C-peptide ELISA kit, Millipore, EZHCP-20K).

**Propium iodide cell viability staining.** Cells at D30_L were dissociated using Accutase and replated onto 96-well plates coated with 804-G conditioned medium. Once attached, cells were stained by 2.5 µg mL$^{-1}$ propium iodide in DMEM for 12 min, washed once with DMEM and fixed with 4% paraformaldehyde for 20 min. DAPI was used to quantify total number of cells after fixation. Plates were analyzed using a Molecular Devices ImageXpress High-Content Analysis System. Data were quantified using MetaXpress software.

**High throughput chemical screening.** To perform the high throughput small-molecule screening, $GLIS3^{-/-}$ cells at D26_L were dissociated using Accutase and replated onto 804G-coated 384-well plates at 20,000 cells/80 µL medium/well. After 8 h, cells were treated at 1 µM and 10 µM with compounds from an in-house library of ~300 signaling pathway modulators, an epigenetics library (Cayman Chemical), Prestwick library of approved drugs (FDA, EMA, and other agencies), LOPAC (Sigma Aldrich) and the MicroSource library totaling ~5000 chemicals. DMSO treatment was used as a negative control. Untreated wells containing WT cells were included as positive control. After 4 days of culture, cells were first stained with 2.5 µg mL$^{-1}$ PI and then fixed and stained using antibodies against Insulin (DAKO) and cleaved caspase-3 (BD biosciences). Plates were analyzed using a Molecular Devices ImageXpress High-Content Analysis System. Two-dimensional analysis was used. Compounds inducing lower % cleaved caspase-3 ($Z$-score $<-1.5$) and a similar or higher number of INS$^+$ cells compared to DMSO treated wells were selected as primary hits.

**Quantitative real-time PCR analysis.** Total RNA was isolated using the Qiagen RNeasy Plus mini kit following manufacturer's instructions. First strand cDNA was generated using the Superscript III FirstStrand Synthesis System (ThermoFisher Scientific). First strand cDNA products were used as qPCR templates in SYBR Green-based qPCR using a Roche 480 Lightcycler. Triplicate reactions (technical replicates) were carried out for each biological replicate. *ACTB* was used as a housekeeping control to normalize target gene expression. Sequences of primers used are listed in Supplementary Table 2.

**Purification of human β-cells from islets for RNA-seq.** Human islets were provided by the IIDP (Integrated Islet Distribution Program). Briefly, 10,000 islets from a healthy donor were partially disaggregated using 0.25% Trypsin/EDTA (Corning), resuspended in RPMI 1640 supplemented with 10% FBS, 1× GlutaMax, 100 U mL$^{-1}$ penicillin/100 µg mL$^{-1}$ streptomycin (GIBCO) and infected with an insulin reporter adenovirus construct pAd-RIP-Zsgreen. Four days later, the cells were dissociated with 0.25% Trypsin/EDTA. Zsgreen$^+$ DAPI$^-$ cells were FACS sorted directly into Trizol LS (ThermoFisher Scientific). RNA was extracted according to the manufacturer's instructions.

**RNA-seq.** Sample QC analysis, cDNA library synthesis, and RNA sequencing were carried out by the Weill Cornell Genomics Core. In brief, the quality of RNA samples was examined by Agilent bioanalyzer (Agilent). cDNA libraries were generated using TruSeq RNA Sample Preparation (Illumina). Each library was sequenced using single-reads in HiSeq4000 (Illumina). Gene expression levels were analyzed using Cufflinks.

**Bioinformatics analysis.** To generate a heatmap plot on three or more samples, the expression values were normalized per gene over all samples. For each gene, we calculated the mean and standard deviation (stdev) of expression over all samples, and linearly transformed the expression value using the formula (RPKM-mean)/stdev. The heatmaps were then generated using heatmap.2 in the R gplots package. GSEA was performed using GSEA software (Broad Institute). To compare PP1-β and PP2-β cells with adult β-cells, gene sets listing the top 1000 genes differentially expressed between PP1-β and adult β-cells were used (UP for genes higher expressed and DN for genes lower expressed in adult β-cells). To create a gene set to analyze the endocrine gene expression signature in WT and $GLIS3^{-/-}$ PP2 cells, genes enriched in NGN3-GFP$^+$ cells from e15.5 mouse pancreas were used (http://discovery.lifemapsc.com/in-vivo-development/pancreas/dorsal-pancreatic-bud/endocrine-progenitor-cells). GO and KEGG pathway analysis on up/down-regulated genes in WT and $GLIS3^{-/-}$ cells were performed using DAVID v6.8 functional annotation tool (https://david.ncifcrf.gov/). Pathways prediction with IPA (QIAGEN Bioinformatics) was carried out using as the input genes up/downregulated ≥2-fold in WT and $GLIS3^{-/-}$ GFP$^+$ PP2-β cells.

**In vivo transplantation and drug treatment.** WT and isogenic mutant hESCs at day 24 of differentiation (around 1 million cells) were harvested by cell scraper, mixed with 20 µl Matrigel (Corning), and transplanted under the kidney capsule of 6–8-week-old male SCID-beige mice. For drug treatment, mice were injected intraperitoneally with 15 mg kg$^{-1}$ day$^{-1}$ galunisertib or vehicle for 7 days. To prepare the injection cocktail, 200 mM galunisertib in DMSO was diluted ~15 times with a 50:50 mixture of PEG300 (Sigma Aldrich) and saline (APP Pharmaceuticals). All animal work was conducted in agreement with NIH guidelines and approved by the local Institutional Animal Care and Use Committee (IACUC), the Institutional Biosafety Committee (IBC) as well as the Embryonic Stem Cell Research Committee (ESCRO).

**Immunohistochemistry.** Mouse kidneys with cells grafted under the capsules were washed with PBS, fixed with 4% paraformaldehyde at 4 °C overnight, and transferred to 30% sucrose solution for dehydration. The tissues were embedded in a 2:1 mixture of OCT: 30% sucrose and sectioned using a cryostat microtome. The slides were blocked and permeabilized in PBS solution containing 5% horse serum and 0.3% Triton for 1 h at RT and then incubated with primary antibodies overnight at 4 °C followed by 1 h incubation with fluorescence-conjugated secondary antibodies (Alexafluor, ThermoFisher Scientific) at RT. The following primary antibodies were used: anti-PDX1 (1:500, R&D), anti-insulin (1:500, DAKO) and anti-cleaved caspase-3 (1:1000, BD Biosciences), and anti-STEM121 (1:1000, Stem Cells Inc.). Fluorescent images were scored using MetaMorph® image analysis software (Molecular Devices).

**Western blot analysis.** Whole-cell lysates were generated by scraping cultures on day 24 of differentiation in cold PBS, and re-suspending in complete lysis buffer (20 mM Tris pH 7.0, 150 mM NaCl, 50 mM NaF, 1% NP-40 substitute, and Thermo Scientific HALT protease inhibitor cocktail 1:100). Lysates were loaded onto 10% NuPage Bis-Tris gels (Invitrogen), resolved by electrophoresis, and transferred to PVDF membranes (Bio-Rad). Membranes were blocked with 5% bovine serum albumin in TBS + 0.05% Tween and probed overnight with primary antibody. The antibodies were rabbit anti-phospho-SMAD2/3 (1:250, Cell Signaling), rabbit anti-SMAD2/3 XP (1:5000, Cell Signaling), and mouse anti-β-actin (1:50,000, Sigma A1978). Membranes were washed and incubated for 1 h with HRP-conjugated secondary antibody (Bio-Rad) in 5% milk-TBS-0.05% Tween and developed using SuperSignal West Pico (Thermo Scientific) or Immobilon (Millipore) ECL substrate.

**Statistical analysis.** Data are presented as mean ± SEM derived from at least three independent biological replicates. Data on biological replicates ($n$) and the type of statistical test are described in the figure legends. Statistical analysis was performed using GraphPad Prism 6 software. $P$ values by unpaired two-tailed $t$-test were *$P < 0.05$, **$P < 0.01$, ***$P < 0.001$, ****$P < 0.0001$; n.s. not significant.

**Data availability.** The authors declare that all data supporting the findings of this study are available within the article and its Supplementary Information Files or from the corresponding author upon reasonable request. All datasets described in the paper have been deposited in NCBI Gene Expression Omnibus under accession number GSE114051.

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

## Acknowledgements

This work was supported by NIDDK (DP2DK098093-01, DP3DK111907-01, 1R01DK116075-01A1) and American Diabetes Association (1–12-JF-06 and 1–17-IBS-019) to S.C. and NIDDK (R01DK096239) to D.H. This study was also supported by shared facility contracts from the New York State Department of Health to T.E., D.H., and S.C. (NYSTEM C029156) and to M.T. (NYSTEM C029153). In addition, this study was supported by Integrated Islet Distribution Program-Beckmann Research Center -City of Hope NIH/NIDDK (to A.N.) and 1S10 OD019986-01 (Weill Cornell). Adenovirus containing rat insulin promoter-driven ZsGreen was kindly provided by Drs. Andrew Stewart and Karen Takane from the Icahn School of Medicine at Mount Sinai. Mouse anti-NKX6.1 and anti-NKX2.2 antibodies were purchased from the Developmental Studies Hybridoma Bank at the University of Iowa. We are also very grateful for technical support and advice provided by Harold S. Ralph in the Cell Screening Core Facility. We would like to thank Faranak Fattahi for valuable comments on the manuscript.

## Author contributions

S.A.: Design and conception of the study, performing experiments, and manuscript preparation; B.C., Ti.Z., Z.G., R.L., M.K., Tu.Z., J.Z.X., M.C., M.P., M.T., Z.Z., C.L., A.N. performed additional experiments; T.E., D.H., S.C.: design and conception of the study, data interpretation, and manuscript preparation.

## Additional information

**Competing interests:** The authors have filed a patent of the differentiation strategy. The authors declare no other competing interests.

