## [Peer Review File · Nature Communications]

Reviewers' comments:

Reviewer #1 (Remarks to the Author):

In this study, the investigators examine the role of GLIS3 in pancreatic beta cell differentiation using human PSCs. They reveal a role for GLIS3 in the regulation of apoptosis and SMAD signaling. Overall the findings are interesting and novel.

1. Abstract, line 8-9: "This phenotype is distinct from Glis3^{-/-} mice, emphasizing the significance of a human disease model". It is a bit premature to conclude this. It is difficult to compare an in vitro cell system with the differentiation of beta cells in vivo where cell environment, e.g., mesenchymal cells, endothelial cells as well as hormones produced by other tissues, can influence this differentiation. Previous data by the same investigators (Ref 20) showed a different result with Glis3 KO hPSCs on cell differentiation than the current study indicating that the differentiation protocol used can greatly influence what effect loss of GLIS3 expression can have on beta cell differentiation. This is another indication that one needs to be cautious translating in vitro data to the phenotype observed in Glis3 deficiency in human or mice. In addition, in humans with GLIS3 deficiency all pancreatic endocrine cells are reduced different from what is observed in the in vitro cell system described here.

2. P4, line 7 in Results: "comprising a pool that contains around 75-90% PDX1⁺ cells (Supplementary Fig. 1a, 1b). I do not see the % listed in these figures.

3. P4, Line 11-12: "However, the derived primary transition PP1- β cells are poly-hormonal (Fig. 1d-1f)" appears to refer to the old differentiation protocol, while in the middle of P5 it is referred to it again "In contrast, only 30-40% of INS⁺ PP1- β cells are mono-hormonal Fig. 1d-1f" for the new differentiation protocol. This is confusing and needs to be corrected.

4. Fig. 1g: the % of UNC3, NKX6.1, PAX6 and Isl1 positive cell should be provided. It is difficult to see from the IHS what % is positive.

5. The relative percentage of Sst⁺ and Ghrl⁺ cell is increased. This relative increase appears to be due to the loss of cells by apoptosis, thus fewer cells, not to an increase in the absolute number of these cells due to increased Sst and Ghrl differentiation. The authors should comment on that in the paper.

6. P 9: The investigators state that most of the Glis3⁻ PP2⁻ cells are PP2 cells, but they also state that Glis3⁻ PP2⁻ cells are more prone to undergo apoptosis. Also seems that in the absence of Glis3 at least some PP2 cells still are able to differentiate into PP2⁻ cells? The PP2 and PP2⁻ terminology should be used for phenotype of the cell not the presumed stage of differentiation. Not clear how they distinguish Glis3⁻ PP2 from Glis3⁻ PP2⁻ cells and what causes the increase in sensitivity to undergo apoptosis in PP2⁻ cells.

7. It is not clear whether the reduced INS gene expression in the PP2⁻ cells is due reduced number of INS⁺ cells or whether INS transcription is affected as well by the lack of GLIS3 expression. Since NGN3 has been reported to be a GLIS3 target gene, I was also wondering whether the lack of GLIS3 affects the number of NGN3⁺ cells. These two questions should be determined and included in the paper.

8. I am surprised about the poor separation between Annexin V⁺ and Annexin V⁻ cells. There appears not to be a clear distinction between the two cell populations. This makes calculation of the number of apoptotic cells less convincing. Can this be improved for example with double staining for ToPro3 and Annexin V as was done in Fig. 3b? This gives a much better separation between apoptotic and nonapoptotic cells.

9. Fig. 5c and d: How do the investigators know that the increase in p-SMAD in PP2⁻ cells is due to changes in INS⁺ cells and not to a shift in the cell populations (reduced percentage of INS⁺ cells). Although the inhibitor hints at that its is related to cells, it also could act on other endocrine cells that via paracrine factors affect cells. It would be good to examine whether p-SMAD co-localize with INS⁺ cells by immunohistochemical staining or flow cytometry.

10. It is not clear how p-SMAD is activated. Is the expression of any of known activators upstream of the SMAD pathway increased in GLIS3 deficient cells?

11. Much of Fig. 5F is pure speculation or just well known facts about the SMAD signaling pathway. I do not see any reason to include this schematic in the main section of the manuscript. It should be deleted or placed in Supplements.

Minor comments;

1. P5, line 12: Figure 1G change to Figure 1g. Also "Figure" and "Fig." are not used consistently.
2. Fig. 1a and Supplemental Fig. 1a are redundant. Delete one of the two Figures.
3. M&M, Immunohistochemistry: correct typo in "BD Biosciences"
4. Figure 3b: ToPro3 misspelled on y-axis.

Reviewer #2 (Remarks to the Author):

Amin et al. develop an extension on published human pluripotent stem cells differentiation protocols, which produces mono-hormonal beta-like cells (referred to as PP2-beta) that are more mature than those achieved through older versions. (Importantly, their PP2-beta cells are glucose responsive, and express maturation markers that are not detected in cells differentiated by the older versions of the protocols). This, by itself, is an important achievement. The authors then use this improved protocol to unmask the role of GLIS3 in human pancreatic development, using a clever isogenic HESC lines analysis. GLIS3 is one of the most important GWAS candidates for diabetes, implicated in both T1D and T2D, and thus most likely to affect beta cells. The authors confirm that GLIS3 is involved in cell death, and use high-content chemical screen to identify the drug galunisertib as a GLIS3-specific candidate for rescuing cell death in GLIS3 mutant cells.

This is an elegant work which maximizes the strengths of human pluripotent stem cell differentiation for disease modeling and drug discovery, uses this platform exactly how it should be used, and reports on several important advancements to the field. I have no major concerns with the manuscript.

Minor points:

- 1) The loading controls for the Western blots in figure 5c and 5d seem way overexposed, and it is hard to conclude anything from the experiment.
- 2) The authors state that "compared with PP1, PP2 cells express higher levels of trunk PP markers, including for PDX1, NKX6.1 and NEUROD1 as indicated by qRT-PCR assays", but the referred figure (figure 1b) shows no significant difference in PDX1 expression between PP1 and PP2 cells.
- 3) Is there significance for the difference in GHRL expression between replicates of PP1-beta expression levels in figure 1h? Are these biological replicates?

Reviewer #3 (Remarks to the Author):

Human embryonic stem cells (hESCs) have recently been used to study the function of genes involved in human embryonic and foetal development. In this article, the authors generate new inactivating alleles of GLIS3 in hESCs and show that they decrease beta cell development. GLIS3 is a gene which causes neonatal diabetes upon homozygous or compound heterozygous mutations and for which heterozygous variants predispose to both type 1 and type 2 diabetes. While inactivation in mice has clarified its role in beta cell development its role in human is only extrapolated from mouse data. The authors convincingly show that GLIS3 reduces beta cell

development and increases cell death in progenitors (only those obtained using specific protocols) and beta cells. The article further shows that it is then possible to screen for drugs that correct the defect and identify TGFb inhibitors. The screen adds tremendously to the study and could inspire others. It is unlikely that it will lead to a treatment using TGFb inhibitors during pregnancy but it clarifies the disease mechanisms. The authors actually end their paper by a note on how this should help in neonatal diabetes treatment and should clarify what they have in mind.

While the article fails to decipher the mechanisms by which GLIS3 controls the TGFb pathway, this is nevertheless a beautiful and comprehensive study. Some requests for improvement are provided below. A major one is whether death is a primary defect or not. Since it occurs rather late in the protocol, it may be a consequence of overcrowding. Is the proliferation rate of PP2 cells the same in WT and GLIS3 KO? It is also striking that it starts two days after the change of medium triggering differentiation. Since death also seems to affect PP2 progenitors prior to their differentiation wouldn't those be expected to die as soon as they become GLIS3 high? This should occur at the latest at 23 days. There is a big gap between day 9 and 23 during which PP1 transform into PP2 (or are selected). Why does it take that long? When does GLIS3 go up in this period?

The second main concern is the lack of focus on NEUROG3 (see more specific comments below). It is important to provide a point of comparison with the data in the knock-out mice. This is all the more important that there is little in common between the phenotype described in mice and in human ESCs other than resulting in less beta cells. If in addition the tools and readout used are different, it limits even more the comparison.

Points needing clarification, improvement or correction:

- It is often unclear in the paper which of the experiments were performed on different KO lines and when done on one, which one was used.
- Abstract, lines 3-4: Was GLIS3 really not expressed in the previous protocol? It is said later that it is, though at lower levels. The reason for the lack of phenotype in previous attempts is rather speculative. The previous mutations were possibly not total loss-of-function. Arguments for the new ones being total loss-of-function should be discussed somewhere.
- Abstract, lines 5: it does not seem justified to talk about a secondary transition in human. This is a concept in mice where there are two waves of endocrine cell production but this does not seem to be conserved in human. The cells may indeed correspond to a later type of progenitor (refer to what it corresponds to in vivo and what the criteria are) but should not be called secondary transition.
- Page 4, line 12: Clarify if it is known when GLIS 3 is expressed in the human pancreas. Do the PP1 cells express GLIS3? This is clarified later but a reader can wonder already at this point.
- Page 5, line 8: Why do the authors write that there is spontaneous differentiation? It is triggered by a medium change isn't it?
- Page 5, line 10: Clarify if this protocol also leads to the differentiation of other monohormonal cell types.
- Page 5, line 15: how do the levels of UNC3 or MAFA compare to mature beta cells? The maturity of beta cells in vitro is a big issue at the moment. It is not the main point of the paper but many readers will be interested to know.
- Page 7, bottom: indicate in the text the effect on GCG cells.
- Page 13, lines 12-14: While the difference with mouse studies may highlight a species difference indeed, it may also reveal a difference between in vitro and in vivo systems. The authors should be more careful.
- Figure 1C: How many PP1 and PP2 samples were used? This looks like n=1 of each, which is not sufficient considering the modest fold changes. The fold changes are at odds with Fig 1B for the genes that could be compared.
- Figure 1K: The ratio of secretion at 20mM glucose versus 2mM is not fantastic for the PP2-b (and for the control islets themselves)

- Figure 2b: repeat numbers? Statistical significance? NEUROG3 seems down. Why are endocrine cell numbers normal??? Is it a reduction in transcript per cell or cells expressing NEUROG3? What happens at the protein level? ChromograninA also seems down though cell numbers are normal. Are these changes in NEUROG3, and CHRGA stable with time in vitro?
- Figure 3b and d: the Annexin V flow profiles are very unusual. Annexin V is usually much easier to gate, forming two different populations rather than the continuum used here. The continuum is much more difficult to gate and could lead to wrong interpretations. Attention to Topro misspelling in b. Cell death is however shown in many different ways and is likely real but since it was not seen in the mouse model that's something to be careful about.
- Figure S1b: Is it not PP2 at day 23?

We would like to thank all the reviewers for the valuable comments on our manuscript and for appreciating the significance and novelty of our work. We understand the reviewers' concerns and have performed additional experiments to address them. These experiments are included as 15 main and 26 supplementary new or modified figure panels and 2 new supplementary tables. The responses to reviewers' comments are detailed as below, highlighted in blue.

Reviewers' comments:

Reviewer #1 (Remarks to the Author):

In this study, the investigators examine the role of GLIS3 in pancreatic beta cell differentiation using human PSCs. They reveal a role for GLIS3 in the regulation of apoptosis and SMAD signaling. Overall the findings are interesting and novel.

1. Abstract, line 8-9: "This phenotype is distinct from Glis3^{-/-} mice, emphasizing the significance of a human disease model". It is a bit premature to conclude this. It is difficult to compare an *in vitro* cell system with the differentiation of beta cells *in vivo* where cell environment, e.g., mesenchymal cells, endothelial cells as well as hormones produced by other tissues, can influence this differentiation. Previous data by the same investigators (Ref 20) showed a different result with Glis3 KO hPSCs on cell differentiation than the current study indicating that the differentiation protocol used can greatly influence what effect loss of GLIS3 expression can have on beta cell differentiation. This is another indication that one needs to be cautious translating *in vitro* data to the phenotype observed in Glis3 deficiency in human or mice. In addition, in humans with GLIS3 deficiency all pancreatic endocrine cells are reduced different from what is observed in the *in vitro* cell system described here.

Response: We understand the reviewer's concern. To address this issue, we have removed the claim from the abstract. In addition, we have added the following description in the discussion. "The difference between *GLIS3*^{-/-} hESC-derived cells and *Glis3*^{-/-} mice might underline the distinction between mouse development and human cell-based systems or the difference between *in vitro* or *in vivo* conditions." at Page 15 Line 1.

In reference to humans with GLIS3 deficiency, we are not aware of any studies that directly analyze the pancreatic tissue of the patients carrying loss of function mutations of *GLIS3*. We would be happy to include the reference if reviewer recommends it.

2. P4, line 7 in Results: "comprising a pool that contains around 75-90% PDX1⁺ cells (Supplementary Fig. 1a, 1b). I do not see the % listed in these figures.

Response: The flow cytometry plot for PDX1⁺ cells has been added as **Supplementary Fig. 1a**. The percentage of PDX1⁺ cells is 83.9 ± 9.1%.

3. P4, Line 11-12: "However, the derived primary transition PP1-β cells are poly-hormonal (Fig. 1d-1f)" appears to refer to the old differentiation protocol, while in the middle of P5 it is referred to it again "In contrast, only 30-40% of INS⁺ PP1-β cells are mono-hormonal Fig. 1d-1f" for the new differentiation protocol. This is confusing and needs to be corrected.

Response: We have changed the description to “the derived primary transition PP1-β cells are mostly poly-hormonal (comprising a population of 60-70% poly-hormonal and 30-40% mono-hormonal INS⁺ cells), which represent the cells from older protocols.” at Page 4 Line 20. In contrast, 85-95% of PP2-β INS⁺ cells derived using new differentiation protocol, are mono-hormonal.

4. Fig. 1g: the % of UNC3, NKX6.1, PAX6 and Isl1 positive cell should be provided. It is difficult to see from the IHS what % is positive.

Response: We have added the percentage of UCN3, NKX6.1, PAX6 and ISL1 positive cells in the main text at Page 5 Lines 16-17.

5. The relative percentage of Sst⁺ and Ghrl⁺ cell is increased. This relative increase appears to be due to the loss of cells by apoptosis, thus fewer cells, not to an increase in the absolute number of these cells due to increased Sst and Ghrl differentiation. The authors should comment on that in the paper.

Response: To address this concern, we quantified the total number of SST⁺ and GHRL⁺ cells in WT and *GLIS3*^{-/-} hESC-derived cells at D30_L which were differentiated at similar starting density. The total number of GHRL⁺ and SST⁺ cells were significantly higher in *GLIS3*^{-/-} cells compared to WT cells. We have included the data in **Supplementary Fig. 4i, 4j**.

6. P 9: The investigators state that most of the Glis3- PP2- cells are PP2 cells, but they also state that Glis3- PP2- cells are more prone to undergo apoptosis. Also seems that in the absence of Glis3 at least some PP2 cells still are able to differentiate into PP2- cells? The PP2 and PP2- terminology should be used for phenotype of the cell not the presumed stage of differentiation. Not clear how they distinguish Glis3- PP2 from Glis3- PP2- cells and what causes the increase in sensitivity to undergo apoptosis in PP2- cells.

Response: Some characters are missing in the reviewer’s comments. We assume that the reviewer means “PP2-β cells” by “PP2- cells”.

First, we agree with the reviewer that it is confusing to use PP2 and PP2-β for both cells and stages. To clarify this issue, the differentiation stages and cells were defined as below and added as the **Supplementary Table 1**.

Stage	Cells
D9/day 9	PP1 cells
D16_E/day 16 using the early progenitor protocol	PP2 cells
D23_L/day 23 using the late progenitor protocol	INS-GFP ⁺ cells were defined as PP1-β cells
D30_L/day 23 using the late progenitor protocol	INS-GFP ⁺ cells were defined as PP2-β cells

Second, reviewer is correct. Although the percentage of INS-GFP⁺ cells decreases at D30_L, loss of GLIS3 does not completely block differentiation to PP2-β cells.

Finally, the percentage of apoptotic cells in *GLIS3*^{-/-} PP2 cells are around two-fold higher than that in WT PP2 cells (**Fig. 3a and 3b**), which is comparable to the fold change of the percentage of apoptotic cells in *GLIS3*^{-/-} PP2-β cells versus WT PP2-β cells (**Fig. 3d, 3e**). However, the overall apoptotic rate is higher in PP2-β cells than PP2 cells, this might due to the change of culture medium. PP2 cells were maintained in a relatively rich medium containing EGF and FGF, which facilitate cell survival and self renewal. However, PP2-β cells were maintained in basal medium containing only B27.

7. It is not clear whether the reduced INS gene expression in the PP2- cells is due reduced number of INS⁺ cells or whether INS transcription is affected as well by the lack of GLIS3 expression. Since NGN3 has

been reported to be a *GLIS3* target gene, I was also wondering whether the lack of *GLIS3* affects the number of *NGN3*⁺ cells. These two questions should be determined and included in the paper.

Response: Again, some characters are missing in the reviewer's comments. We assume that reviewer means "PP2-β cells" by "PP2- cells".

To determine whether insulin transcription is affected by the lack of *GLIS3* expression, qRT-PCR analysis was applied to monitor insulin transcriptional expression in the purified WT and *GLIS3*^{-/-} INS-GFP⁺ PP2-β cells. Indeed, the transcriptional expression of insulin in *GLIS3*^{-/-} INS-GFP⁺ PP2-β cells is significantly lower than that of WT INS-GFP⁺ PP2-β cells. We have included the data in **Supplementary Fig. 4k**.

Regarding the *NGN3*⁺ cells, immunostaining was used to quantify the number and percentage of *NGN3*⁺ cells in WT or *GLIS3*^{-/-} hESC-derived cells at D9, D16_L and D23_L. We did not observe significant difference between WT or *GLIS3*^{-/-} hESC-derived cells at any of the time points tested. We have included the data in **Supplementary Fig. 3c, 3d**.

8. I am surprised about the poor separation between Annexin V⁺ and Annexin V⁻ cells. There appears not to be a clear distinction between the two cell populations. This makes calculation of the number of apoptotic cells less convincing. Can this be improved for example with double staining for ToPro3 and Annexin V as was done in Fig. 3b? This gives a much better separation between apoptotic and nonapoptotic cells.

Response: We apologize for the poor separation of Annexin V⁺ and Annexin V⁻ cells. To determine whether this is due to hESC-derived population or the staining protocol, we first tested the same staining protocol on EndoC-βH1 cells and found that a clear separation of Annexin V⁺ and Annexin V⁻ cells in EndoC-βH1 cell line. (**a** in the following figure). Secondly, to improve the accuracy of gating strategy, we have included a positive control, the cells treated with 10 μM Camptothecin (Sigma) for 4 hours, to facilitate gating (**c** and **d** in the following figure and **Supplementary Fig. 5c**).

In addition, we repeated experiments in **Fig 3b, 3d** and Supplementary Fig 3a (now **Supplementary Fig. 4c**) with another staining kit with a different fluorescence conjugate (the old kit: PE-annexin V from BD Biosciences and a new kit Alexa 647-conjugated annexin V from Thermo Fisher Scientific). Similarly, cells treated with 10 μM Camptothecin and EndoC-βH1 cells were used as positive controls to facilitate gating. The gate set based on the Camptothecin-treated sample was also applied to the INS-GFP⁺ cells (**Fig. 3d, 3e**).

Together, we have repeated our experiments using two Annexin V staining kits with different fluorescence conjugates. In addition, we have added the positive control of EndoC-βH1 cells and hESC-derived cells treated with Camptothecin to facilitate gating. Importantly, the Annexin V staining using two different staining kits led to the same conclusion that *GLIS3*^{-/-} cells show a significantly increased cell apoptosis compared to WT cells. This can be visualized by plotting the live cells stained for annexin V on a histogram (**Fig. 3f**). We also measured the median fluorescence intensity (MFI) and found that the *GLIS3*^{-/-} cells have significantly higher MFI for Annexin V compared to WT cells (**Fig. 3g**). We feel these data should be sufficient to support our conclusion that *GLIS3*^{-/-} cells show a significantly increased cell apoptosis than WT cells.

Figure 1. Optimization of Annexin V staining to measure apoptosis. (a, b) Annexin V staining on EndoC-βH1 (a) and PP2-β (b) cells using PE-conjugated Annexin V (BD Biosciences) vs A647-conjugated Annexin V (Thermo Fisher Scientific) dyes. (c, d) Annexin V staining on EndoC-βH1 (a) and PP2-β (b) cells following treatment with 10 μM Camptothecin vs DMSO for 4 hours.

9. Fig. 5c and d: How do the investigators know that the increase in p-SMAD in PP2- cells is due to changes in INS⁺ cells and not to a shift in the cell populations (reduced percentage of INS⁺ cells). Although the inhibitor hints at that its is related to cells, it also could act on other endocrine cells that via paracrine factors affect cells. It would be good to examine whether p-SMAD co-localize with INS⁺ cells by immunohistochemical staining or flow cytometry.

Response: To address the reviewer's concern, we have performed staining of p-SMAD2/3. Indeed, nuclear staining of p-SMAD2/3 was detected in *GLIS3*^{-/-} cells at D30_L, but not WT cell at D30_L. In addition, all INS⁺ cells have nuclear expression of p-SMAD2/3. Interestingly, the increase in p-SMAD2/3 was observed in both INS⁺ cells and INS⁻ cells. The image has been added as **Supplementary Fig. 9b**. Consistent with this data, the inhibitor rescues cell death in both INS⁺ and INS⁻ cells (**Fig. 4d-4f and Supplementary Fig. 8a, 8b**).

10. It is not clear how p-SMAD is activated. Is the expression of any of known activators upstream of the SMAD pathway increased in *GLIS3* deficient cells?

Response: RNA-seq analysis suggested that both TGFβ ligands (TGFβ2 and TGFβ3) and receptor (TGFBR2) are significantly upregulated in *GLIS3*^{-/-} cells compared with WT cells at D23_L, suggesting

that loss of *GLIS3* might activate p-SMAD signaling by upregulating the above genes. We have added this data in **Supplementary Fig. 9a**.

11. Much of Fig. 5F is pure speculation or just well known facts about the SMAD signaling pathway. I do not see any reason to include this schematic in the main section of the manuscript. It should be deleted or placed in Supplements.

Response: We have moved the old Fig. 5f to Supplementary Fig. 9c.

Minor comments;

1. P5, line 12: Figure 1G change to Figure 1g. Also “Figure” and “Fig.” are not used consistently.

Response: We have changed it to Fig. 1g at page 5, line 14.

2. Fig. 1a and Supplemental Fig. 1a are redundant. Delete one of the two Figures.

Response: We have deleted the schematic in Supplementary Fig. 1a.

3. M&M, Immunohistochemistry: correct typo in “BD Bioscences”

Response: We apologize for the typo. This has been corrected.

4. Figure 3b: ToPro3 misspelled on y-axis.

Response: In the new Fig 3b, we showed new flow cytometry analyses using DAPI.

Reviewer #2 (Remarks to the Author):

Amin et al. develop an extension on published human pluripotent stem cells differentiation protocols, which produces mono-hormonal beta-like cells (referred to as PP2-beta) that are more mature than those achieved through older versions. (Importantly, their PP2-beta cells are glucose responsive, and express maturation markers that are not detected in cells differentiated by the older versions of the protocols). This, by itself, is an important achievement. The authors then use this improved protocol to unmask the role of *GLIS3* in human pancreatic development, using a clever isogenic HESC lines analysis. *GLIS3* is one of the most important GWAS candidates for diabetes, implicated in both T1D and T2D, and thus most likely to affect beta cells. The authors confirm that *GLIS3* is involved in cell death, and use high-content chemical screen to identify the drug galunisertib as a *GLIS3*-specific candidate for rescuing cell death in *GLIS3* mutant cells.

This is an elegant work which maximizes the strengths of human pluripotent stem cell differentiation for disease modeling and drug discovery, uses this platform exactly how it should be used, and reports on several important advancements to the field. I have no major concerns with the manuscript.

Response: We thank the reviewer for acknowledgement of the significance and novelty of the manuscript.

Minor points:

1) The loading controls for the Western blots in figure 5c and 5d seem way overexposed, and it is hard to conclude anything from the experiment.

Response: We have included the lower exposed blots for both experiments. The quantification of Fig. 5c and 5d was not affected since the relative phosphorylation level was normalized to total SMAD2/3, as its signal has a wide linear range of detection. In addition, the increase of p-SMAD2/3 in *GLIS3*^{-/-} cells was further confirmed by immunostaining (**Supplementary Fig. 9b**).

2) The authors state that "compared with PP1, PP2 cells express higher levels of trunk PP markers, including for PDX1, NKX6.1 and NEUROD1 as indicated by qRT-PCR essays", but the referred figure (figure 1b) shows no significant difference in PDX1 expression between PP1 and PP2 cells.

Response: We apologize for the confusion. We have corrected this in the text and changed the sentence to "Compared with PP1, PP2 cells express higher levels of late trunk PP markers, including *NKX6.1* and *NEUROD1* as indicated by qRT-PCR assays" at Page 5, Line 10.

3) Is there significance for the difference in GHRL expression between replicates of PP1-beta expression levels in figure 1h? Are these biological replicates?

Response: While the FPKM values for *GHRL* is higher in both PP1- β samples, it does not reach significance. Overall, our immunostaining and flow cytometry data (**Supplementary Fig. 1i, 1j**) suggest that *GHRL* is not strongly co-expressed with INS in neither PP1- β cells nor PP2- β cells. Thus, no significant difference was detected. In the old version of the manuscript, we included all endocrine markers, including GHRL, in the heatmap for the reader's reference. If the reviewer thinks it is inappropriate, we will remove this gene from the heatmap. The samples shown in **Fig. 1h** are biological replicates.

Reviewer #3 (Remarks to the Author):

Human embryonic stem cells (hESCs) have recently been used to study the function of genes involved in human embryonic and foetal development. In this article, the authors generate new inactivating alleles of *GLIS3* in hESCs and show that they decrease beta cell development. *GLIS3* is a gene which causes neonatal diabetes upon homozygous or compound heterozygous mutations and for which heterozygous variants predispose to both type 1 and type 2 diabetes. While inactivation in mice has clarified its role in beta cell development its role in human is only extrapolated from mouse data. The authors convincingly show that *GLIS3* reduces beta cell development and increases cell death in progenitors (only those obtained using specific protocols) and beta cells. The article further shows that it is then possible to screen for drugs that correct the defect and identify TGF β inhibitors. The screen adds tremendously to the study and could inspire others. It is unlikely that it will lead to a treatment using TGF β inhibitors during pregnancy but it clarifies the disease mechanisms. The authors actually end their paper by a note on how this should help in neonatal diabetes treatment and should clarify what they have in mind.

Response: We thank the reviewer for the appreciation of the significance of the manuscript. We agree with reviewer 3 that it might be difficult to apply TGF β inhibitor during pregnancy. Since many *GLIS3* SNPs are associated strongly with both T1D and T2D, it is conceivable that they share a common mechanism

with our knockout model. The identified drug candidates can potentially be used to treat patients carrying *GLIS3* SNPs that lead to decrease/loss in *GLIS3*. More importantly, the TGF β inhibitor provides a chemical tool to dissect the downstream signal perturbations in the absence of *GLIS3*. We have discussed this issue in the main text at Page 15 Line 21.

While the article fails to decipher the mechanisms by which *GLIS3* controls the TGF β pathway, this is nevertheless a beautiful and comprehensive study. Some requests for improvement are provided below. A major one is whether death is a primary defect or not. Since it occurs rather late in the protocol, it may be a consequence of overcrowding. Is the proliferation rate of PP2 cells the same in WT and *GLIS3* KO?

Response:

1. We performed additional RNA-seq analysis and found TGF β ligands (TGF β 2 and TGF β 3) and receptor (TGFBR2) are significantly upregulated in *GLIS3*^{-/-} cells compared with WT cells at D23_L, suggesting that loss of *GLIS3* might activate p-SMAD signaling by upregulating the above genes. We have added this data in **Supplementary Fig. 9a**. We agree with the reviewer that the detailed mechanism by which *GLIS3* controls the TGF β pathway still needs further investigation.

2. To monitor the proliferation rate, WT and *GLIS3*^{-/-} PP2 cells were stained with Ki67 (**Supplementary Fig. 4b**). There is no significant difference regarding the cell proliferation rate of WT and *GLIS3*^{-/-} PP2 cells. In addition, when WT and *GLIS3*^{-/-} PP2 cells were replated at similar density, *GLIS3*^{-/-} cells still show higher apoptotic rate. Together, it suggests that the high cell death in *GLIS3*^{-/-} cells is not a consequence of overcrowding.

It is also striking that it starts two days after the change of medium triggering differentiation. Since death also seems to affect PP2 progenitors prior to their differentiation wouldn't those be expected to die as soon as they become *GLIS3* high? This should occur at the latest at 23 days. There is a big gap between day 9 and 23 during which PP1 transform into PP2 (or are selected). Why does it take that long? When does *GLIS3* go up in this period?

Response: We monitored the *GLIS3* expression from at D9, D12_L, D16_L, D19_L and D23_L during the PP1 to PP2 transition and found the *GLIS3* was not significantly increased until day 23 (**Supplementary Fig. 1f**). Thus, no strong difference of cell apoptosis between *GLIS3*^{-/-} and WT cells until day 23 (**Fig. 3b, 3c**). The percentage of apoptotic cells in *GLIS3*^{-/-} PP2 cells are around two fold higher than that in WT PP2 cells (**Fig. 3a and 3b**), which is comparable to the fold change of the percentage of apoptotic cells in *GLIS3*^{-/-} PP2- β cells versus WT PP2- β cells (**Fig. 3d and 3e**). However, the overall apoptotic rate is higher in PP2- β cells than PP2 cells, this might due to the change of the culture medium. PP2 cells were maintained in a relatively rich medium containing EGF and FGF, which facilitate cell survival. However, PP2- β cells were maintained in basal medium containing only B27. This emphasizes the importance of using a minimal component differentiation protocol when using hPSC-derived cells to study diseases.

The second main concern is the lack of focus on *NEUROG3* (see more specific comments below). It is important to provide a point of comparison with the data in the knock-out mice. This is all the more important that there is little in common between the phenotype described in mice and in human ESCs other than resulting in less beta cells. If in addition the tools and readout used are different, it limits even more the comparison.

Response: We have performed a new set of experiments to address reviewer's comment on *NEUROG3*, which are detailed in the response to the specific comment below.

In addition, we understand reviewer's concern regarding the description of species difference. To address this issue, we have removed the claim from the abstract. In addition, we have added the following description in the discussion. "The difference between *GLIS3*^{-/-} hESC-derived cells and *Glis3*^{-/-} mice might underline the distinction between mouse development and human cell-based systems or the difference between *in vitro* or *in vivo* conditions.

Points needing clarification, improvement or correction:

1- It is often unclear in the paper which of the experiments were performed on different KO lines and when done on one, which one was used.

Response: We have added the detailed information of WT and *GLIS3*^{-/-} clonal lines used in each figure panel in the **Supplementary Table 7**.

2- Abstract, lines 3-4: Was GLIS3 really not expressed in the previous protocol? It is said later that it is, though at lower levels. The reason for the lack of phenotype in previous attempts is rather speculative. The previous mutations were possibly not total loss-of-function. Arguments for the new ones being total loss-of-function should be discussed somewhere.

Response: We have added the RNA-seq data of GLIS3 expression in INS-GFP⁺ cells derived using the previous protocol. The expression level is <1, which is considered as "not expressed" (**Fig. 1k**). To be more accurate, we have removed the claim that GLIS3 is expressed at low level. We also confirmed the results by qPCR on the unsorted cells differentiated using their protocol and cell line. At the terminal stage of the differentiation (day 17), the level of GLIS3 expression is comparable to that of PP1 cells in our protocol. The data is included in the **Supplementary Fig. 1e**.

The previous study generated lines harboring frameshift mutations in the zinc finger domain of GLIS3. We have added the discussion "We cannot fully exclude the possibility that the previous mutations are not total loss-of-function" at Page 14 Line 14.

3- Abstract, lines 5: it does not seem justified to talk about a secondary transition in human. This is a concept in mice where there are two waves of endocrine cell production but this does not seem to be conserved in human. The cells may indeed correspond to a later type of progenitor (refer to what it corresponds to *in vivo* and what the criteria are) but should not be called secondary transition.

Response: We have change the names to early stage pancreatic progenitor (PP1) and late stage pancreatic progenitor (PP2). We would be open to further discussion if reviewer thinks other names are more appropriate.

4- Page 4, line 12: Clarify if it is known when GLIS 3 is expressed in the human pancreas. Do the PP1 cells express GLIS3? This is clarified later but a reader can wonder already at this point.

Response: GLIS3 is expressed in adult human islets (**Supplementary Fig. 1d**). Due to the ethical concerns, it is very challenging for us to get fetal pancreatic tissue to monitor the GLIS3 expression.

5- Page 5, line 8: Why do the authors write that there is spontaneous differentiation? It is triggered by a medium change isn't it?

Response: The reason for using the term "spontaneous differentiation" was to indicate that factors stimulating/forcing the differentiation are not added to the cells. To avoid confusion, we have changed it to

“differentiation”. We would be open to further discussion if reviewer recommends other appropriate descriptions.

6- Page 5, line 10: Clarify if this protocol also leads to the differentiation of other monohormonal cell types.

Response: The reviewer raises an important question. We have performed additional flow cytometry experiments to address this question and have included the results in the **Supplementary Fig. 1m**. Indeed, the majority of α -like, δ -like and ϵ -like cells that are derived using this protocol are monohormonal.

7- Page 5, line 15: how do the levels of UCN3 or MAFA compare to mature beta cells? The maturity of beta cells in vitro is a big issue at the moment. It is not the main point of the paper but many readers will be interested to know.

Response: We assume that reviewer mean “UCN3” instead of “UNC3”.

qRT-PCR was performed to compare the expression of *UCN3* or *MAFA* in INS-GFP⁺ PP2- β cells and human islets. There is not a statistically significant difference between PP2- β cells and human islets. We indeed detected a big variation of *UCN3* and *MAFA* expression in primary human islets. This data has been added as **Supplementary Fig. 1n**.

8- Page 7, bottom: indicate in the text the effect on GCG cells.

Response: We have now indicated the effect on GCG cells in the paragraph’s conclusion at Page 8 Line 18.

9- Page 13, lines 12-14: While the difference with mouse studies may highlight a species difference indeed, it may also reveal a difference between in vitro and in vivo systems. The authors should be more careful.

Response: We understand the reviewer’s concern. To address this issue, we have removed the claim from the abstract. In addition, we have added the following description in the discussion. “The difference between *GLIS3*^{-/-} hESC-derived cells and *Glis3*^{-/-} mice might underline the distinction between mouse development and human cell-based systems or the difference between *in vitro* or *in vivo* conditions.

10- Figure 1C: How many PP1 and PP2 samples were used? This looks like n=1 of each, which is not sufficient considering the modest fold changes. The fold changes are at odds with Fig 1B for the genes that could be compared.

Response: We have added samples in RNA-seq experiments. In the new Figure 1C, n=2 biological replicates for PP1 and n=3 biological replicates for PP2.

11- Figure 1K: le ratio of secretion at 20mM glucose versus 2mM is not fantastic for the PP2-b (and for the control islets themselves)

Response: We performed additional GSIS experiments. The fold induction of primary human islets varies between 1.09 to 4.51 folds. Although the ratio of GSIS of cells at D30_L derived using current protocol is not comparable to primary adult islets, the major focus of this manuscript is to study the role of GLIS3 in human pancreatic differentiation and hESC-derived β -like cell survival. As the reviewer 3 mentioned before, deriving functional mature human β -cells is not the key point of this manuscript. We feel that PP2 and PP2- β cells derived using current protocol already provide some useful information on the role of GLIS3 in human pancreatic differentiation and hESC-derived β -like cell survival, and facilitate understating of the downstream signal of GLIS3.

In addition, we added the following in the discussion “The GSIS response of PP2-β cells is not indistinguishable from human primary islets. Additional optimization might be required to further increase PP2-β cells’ response to glucose stimulation.” at Page 14 Line 5.

12- Figure 2b: repeat numbers? Statistical significance? NEUROG3 seems down. Why are endocrine cell numbers normal??? Is it a reduction in transcript per cell or cells expressing NEUROG3? What happens at the protein level? ChromograninA also seems down though cell numbers are normal. Are these changes in NEUROG3, and CHRGA stable with time in vitro?

Response: We have included the repeat numbers and statistical analysis in Fig. 2b. If anything is missing, we would be happy to fix them.

In addition, immunostaining was used to quantify the number and percentage of NEUROG3⁺ cells in WT or *GLIS3*^{-/-} hESC-derived cells at D9, D16_L and D23_L. We did not observe a significant difference between WT or *GLIS3*^{-/-} hESC-derived cells at any of the time points tested. We have included the data in **Supplementary Fig. 3c, 3d**. However, the fluorescence intensity of NEUROG3⁺ cells in *GLIS3*^{-/-} population is significantly lower than WT NEUROG3⁺ cells (**Supplementary Fig. 3g-3i**).

We also monitored the percentage of ChromograninA⁺ (CHGA⁺) cells, which is shown as the **Supplementary Fig 3e, 3f**. Consistent with NEUROG3⁺ cells, the number and percentage of CHGA⁺ cells are comparable in WT and *GLIS3*^{-/-} cells.

Together, it suggests that *GLIS3* knockout does not affect the formation of endocrine cells and their progenitors at different stages of differentiation.

13- Figure 3b and d: the Annexin V flow profiles are very unusual. Annexin V is usually much easier to gate, forming two different populations rather than the continuum used here. The continuum is much more difficult to gate and could lead to wrong interpretations. Attention to Topro misspelling in b. Cell death is however shown in many different ways and is likely real but since it was not seen in the mouse model that’s something to be careful about.

Response: We apologize for the poor separation of Annexin V⁺ and Annexin V⁻ cells. To determine whether this is due to hESC-derived population or the staining protocol, we first tested the same staining protocol on EndoC-βH1 cells and found that a clear separation of Annexin V⁺ and Annexin V⁻ cells in EndoC-βH1 cell line. (**a** in the following figure). Secondly, to improve the accuracy of gating strategy, we have included a positive control, cells treated with 10 μM Camptothecin (Sigma) for 4 hours, to facilitate gating (**c** and **d** in the following figure and **Supplementary Fig. 5c**).

In addition, we repeated experiments in **Fig 3b, 3d** and Supplementary Fig 3a (now **Supplementary Fig. 4c**) with another staining kit with a different fluorescence conjugate (the old kit: PE-annexin V from BD Biosciences and a new kit Alexa 647-conjugated annexin V from Thermo Fisher Scientific). Similarly, the cells treated with 10 μM Camptothecin and EndoC-βH1 cells were used as positive controls to facilitate gating. The PP2-β cells did not form distinct populations with the A647 kit either (**a** and **b** in the following figure). The gate set based on the Camptothecin-treated sample was also applied to the INS-GFP⁺ cells (**Fig 3d, 3e**).

Together, we have repeated our experiments using two Annexin V staining kits with different fluorescence conjugates. In addition, we have added the positive control of EndoC-βH1 cells and hESC-derived cells treated with Camptothecin to facilitate gating. Importantly, the Annexin V staining using two different staining kits led to the same conclusion that *GLIS3*^{-/-} cells show a significantly increased cell apoptosis compared to WT cells. This can be visualized by plotting the live cells stained for annexin V on a histogram

(Fig. 3f). We also measured the median fluorescence intensity (MFI) and found that the *GLIS3*^{-/-} cells have significantly higher MFI for Annexin V compared to WT cells (Fig. 3g). We feel these data should be sufficient to support our conclusion that *GLIS3*^{-/-} cells show a significantly increased cell apoptosis than WT cells.

Figure 1. Optimization of Annexin V staining to measure apoptosis. (a, b) Annexin V staining on EndoC-βH1 (a) and PP2-β (b) cells using PE-conjugated Annexin V (BD Biosciences) vs A647-conjugated Annexin V (Thermo Fisher Scientific) dyes. (c, d) Annexin V staining on EndoC-βH1 (a) and PP2-β (b) cells following treatment with 10 μM Camptothecin vs DMSO for 4 hours.

14- Figure S1b: Is it not PP2 at day 23?

We thank the reviewer for catching this mistake. We have now corrected it in the current version.

REVIEWERS' COMMENTS:

Reviewer #1 (Remarks to the Author):

1. It is still clear not why this study comes to a different conclusion than their previous paper published in *Cell Stem Cell*, 18: 755 (2016), in which the same authors describe the generation of *Glis3*^{-/-} hESCs and show that no defects were observed in the generation of pancreatic beta cells using a slightly different multi-step differentiation protocol. This could be clarified better
2. Response from authors to question 5: "we quantified the total number of SST⁺ and GHRL⁺ cells in WT and *GLIS3*^{-/-} hESC-derived cells at D30_L which were differentiated at similar starting density. The total number of GHRL⁺ and SST⁺ cells were significantly higher in *GLIS3*^{-/-} cells compared to WT cells". Does this mean that loss of *GLIS3* directs differentiation of endocrine progenitors along the GHRL cell and SST cell instead of the beta cell lineage?

Reviewer #3 (Remarks to the Author):

The authors have satisfactorily addressed all my comments. This paper is truly outstanding.

REVIEWERS' COMMENTS:

Reviewer #1 (Remarks to the Author):

1. It is still clear not why this study comes to a different conclusion than their previous paper published in Cell Stem Cell, 18: 755 (2016), in which the same authors describe the generation of Glis3^{-/-} hESCs and show that no defects were observed in the generation of pancreatic beta cells using a slightly different multi-step differentiation protocol. This could be clarified better.

Response: The fact that no defect was detected in *GLIS3*^{-/-} hESCs in Cell Stem Cell, 18: 755 (2016) is mainly due to the limitation of the protocol. Compared to PP2 cells, the D17 cells generated using the Cell Stem Cell protocol show limited *GLIS3* expression, which explains why Cell Stem Cell studies did not detect defects of *GLIS3*^{-/-} hESCs²⁰ (**Supplementary Fig. 1e**). In contrast, the PP2 and PP2-β cells derived using our new protocol show high *GLIS3* expression (**Supplementary Fig. 1e and 1f**). We have included this explanation in the text at P6, Line 10.

2. Response from authors to question 5: "we quantified the total number of SST⁺ and GHRL⁺ cells in WT and *GLIS3*^{-/-} hESC-derived cells at D30_L which were differentiated at similar starting density. The total number of GHRL⁺ and SST⁺ cells were significantly higher in *GLIS3*^{-/-} cells compared to WT cells". Does this mean that loss of *GLIS3* directs differentiation of endocrine progenitors along the GHRL cell and SST cell instead of the beta cell lineage?

Response: As the reviewer pointed out, our data suggests that loss of *GLIS3* directs the differentiation of endocrine progenitors along the GHRL⁺ ε-like and SST⁺ δ-like cells instead of INS⁺ β-like cells lineage. We have included this conclusion in the text at P15, Line 6.

Reviewer #3 (Remarks to the Author):

The authors have satisfactorily addressed all my comments. This paper is truly outstanding

Response: We thank Reviewer #3 for the appreciation of our work.